



Dynamics of environmental conditions during a decline of a *Cymodocea nodosa* meadow
Mirjana Najdek[1], Marino Korlević[1], Paolo Paliaga[2], Marsej Markovski[1], Ingrid Ivančić[1],
Ljiljana Iveša[1], Igor Felja[3] and Gerhard J. Herndl[4,5]
[1]Center for Marine Research, Ruđer Bošković Institute, G. Paliaga 5, 52210 Rovinj, Croatia
[2]Department of Natural and Health Sciences, University of Pula, Zagrebačka 30, 52100 Pula,
Croatia
[3]Department of Geology, Faculty of Science, University of Zagreb, Horvatovac 102a, 10000
Zagreb, Croatia
[4]Limnology and Bio-Oceanography, Center of Functional Ecology, University of Vienna,
Althanstrasse 14, 1090 Vienna, Austria
[5]NIOZ, Department of Marine Microbiology and Biogeochemistry, Royal Netherlands
Institute for Sea Research, Utrecht University, PO Box 59, Alberta, Den Burg, 1790, The
Netherlands
*Correspondence to*: Mirjana Najdek (najdek@cim.irb.hr)

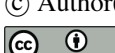



**Abstract.** The dynamics of the physicochemical and biological parameters were followed
during the decline of a *Cymodocea nodosa* meadow in the northern Adriatic Sea from July
2017 to October 2018. During the regular growth of *C. nodosa* from July 2017 to March
2018, *C. nodosa* successfully adapted to the changes of environmental conditions and
prevented $H_2S$ accumulation by its re-oxidation, supplying the sediment with $O_2$ from the
water column and/or leaf photosynthesis. The *C. nodosa* decline was most likely triggered in
April 2018 by a reduction of light availability which affected photosynthesis of *C. nodosa* and
the oxidation capability of below-ground tissue. Simultaneously, a depletion of oxygen due to
intense oxidation of $H_2S$ occurred in the sediment, thus creating anoxic conditions in most of
the rooted areas. These linked negative effects on the plant performance caused an
accumulation of $H_2S$ in the sediments of the *C. nodosa* meadow. During the decay of above-
and below-ground tissues, culminating in August 2018, high concentrations of $H_2S$ were
reached and accumulated in the sediment as well as in bottom waters. The influx of
oxygenated waters in September 2018 led to the re-establishment of $H_2S$ oxidation and
recovery of the below-ground tissue. Our results indicate that if disturbance of environmental
conditions, particularly those compromising the light availability, takes place during the
recruitment phase of plant growth when metabolic needs are at maximum and stored reserves
minimal, a sudden and drastic decline of the seagrass meadow occurs.





## 1 Introduction

Seagrasses are important ecosystem engineers constructing valuable coastal habitats which play a key role in the preservation of marine biodiversity and carbon sequestration (Duarte et al., 2005). Seagrasses extend their active metabolic surfaces (i.e., leaves, rhizomes and roots) into the water column and in the sediment, where root activity might modify the chemical conditions (Marbà and Duarte, 2001). Their canopies and dense meadows are responsible for trapping substantial amounts of sediment particles and organic matter enhancing water transparency and sediment stability with the dense network formed by the rhizome (Gacia and Duarte, 2001; Hendriks et al., 2008; Widdows et al., 2008). Seagrass rhizospheres store organic matter (Pedersen et al., 1997), promote sulfate reduction (Holmer and Nielsen, 1997), release oxygen (Pedersen et al., 1998) and alter sediment redox potential.

Seagrasses require some of the highest levels of solar radiation of any plant worldwide to provide oxygen to roots and rhizomes and support a large amount of non-photosynthetic tissue (Orth et al., 2006). These high solar radiation requirements make seagrasses sensitive to environmental changes, especially those that deteriorate light availability, such as sediment loading, eutrophication or epiphyte cover on seagrass leaves (Terrados et al., 1998; Halun et al., 2002; Brodersen et al., 2015; Costa et al., 2015). Seagrasses have adapted to a highly variable light environment providing tolerance to short-term periods of low light conditions by balancing carbon supply and respiratory requirements. In a healthy growing population this balance is achieved by increasing the photosynthetic activity, re-allocation of carbohydrate reserves from rhizomes and slowing down growth rates (Collier et al., 2009). Beside metabolic and physiological changes, stress responses under poor light conditions include shedding of leaves and shoots and production of new, altered tissue. At sub-lethal light levels, these changes may be permanent. Below these species-specific minimum light requirements seagrass populations are dying off (Collier et al., 2012). Membrane lipids, particularly polyunsaturated fatty acids (PUFA), as the most responsive constituents have a major role in the adaptation processes of primary producers to fluctuating environmental factors, such as temperature, irradiance or salinity (Viso et al., 1993; Lee et al., 2007; Schmid et al., 2014; Sousa et al., 2017; Beca-Carretero et al., 2018; Beca-Carretero et al., 2019). The changes in the unsaturation degree (UND) of membrane fatty acids affect the maintenance of membrane functions and its resistance to cold stress or poor light conditions. UND depends mostly on the variation of α-linolenic (C18:3n-3, ALA) and linoleic (C18:2n-6, LA), the major unsaturated fatty acids in leaves, implicated in the evolution of oxygen during photosynthesis.



LA and ALA are derived from oleic acid by desaturation in the chloroplast and this
conversion considerably declines in the dark, being completely inhibited by anaerobiosis
(Harris and James, 1965).
Sediments inhabited by seagrasses are usually anoxic, highly reduced and rich in sulfide
($H_2S$), a strong phytotoxin (Koch and Erskine, 2001) which has been implicated in several
die-off events of seagrasses (Carlson et al., 1994; Borum et al., 2005; Krause-Jensen et al.,
2011). $H_2S$ is produced by sulfate-reducing bacteria that use sulfate as a terminal electron
acceptor for the mineralization of organic matter (Jørgensen, 1977; Capone and Kiene, 1988,
Canfield et al., 1993). High $H_2S$ concentrations may occur as a consequence of enhanced
mineralization due to increased temperature, organic loading or oxygen depletion (Moeslund
et al., 1994; Pérez et al., 2007; Mascaró et al., 2009). Under these conditions, sulfides may
intrude into plant. Re-oxidation of $H_2S$ in the rhizosphere by incorporation of $S^0$ in the below-
ground tissue has been recognized as a major survival strategy of seagrasses in sulfidic
sediments (Pedersen et al., 2004; Holmer et al., 2005; Hasler-Sheetal and Holmer, 2015).
Generally, the synergistic effect of oxygen depletion and other stresses, such as sulfide
toxicity may shorter the survival of benthic communities and possibly accelerate mortality
events (Vaquer-Sunyer and Duarte, 2010).
The seagrass *Cymodocea nodosa* (Ucria) Ascherson is a common species throughout the
Mediterranean, adapted to a wide range of coastal habitats and environmental conditions
(Terrados and Ros, 1992; Marbà et al., 1996; Pedersen et al., 1997; Zavodnik et al., 1998;
Cancemi et al., 2002; Agostini et al., 2003). During this study, performed from July 2017 to
October 2018 in Saline Bay (northern Adriatic Sea), a considerable decline of *C. nodosa*
meadow occurred. We conducted a series of monthly physicochemical and biological
measurements in *C. nodosa* tissues, sediment underlying the *C. nodosa* meadow, non-
vegetated sediments and surrounding water to i) determine the link between ambient seawater
and sediment environmental factors influencing the growth of *C. nodosa*, ii) document the
response of *C. nodosa* to the changes in environmental conditions that led to the meadow
decline and iii) evaluate the conditions leading to the decline of *C. nodosa*.

**2  Materials and methods**
2.1 Study site
Saline Bay is located 4 km northwest of Rovinj (Croatia) at the coast of the northern Adriatic
Sea (45°7′5″N; 13°37′20″E). The bay represents the terminal shallow part of an 800 m long
inlet, open towards the northwest. The southeastern coast of Saline Bay is characterized by



relatively pristine conditions, while the northwestern littoral part has been completely
modified by the excavation of coastal mud and the addition of large amounts of gravel to
create an artificial beach. Large amounts of silty red soil (*terra rossa*) can be found in the
south eastern inner part of the bay in a large muddy flatland which is slowly being eroded by
the sea and rain weathering. The main input of freshwater to the bay represents land drainage
canals since the year 2017. Even though Saline Bay is protected from the prevailing winds
(from the NE and SE) circulations from the northwestern quadrant can occasionally trigger
bigger waves resuspending the surface sediments and giving the waters a muddy appearance.
A monthly field survey carried from July 2017 to October 2018 has revealed a substantial
decline of *C. nodosa.* At the beginning of this study, the seafloor was covered with large
meadows spreading from the southwestern coastal area (1.5 m depth) toward the central part
of the bay (4 m depth), while at the end of the study only a few small patches persisted in tiny
stripes along the shoreline.

2.2 Sampling
Seawater for analyses of nutrients, chlorophyll a (Chl *a*), particulate matter concentration and
prokaryotic abundance was sampled using plastic containers (10 L). *C. nodosa* was collected
together with rhizomes, roots and epiphytic macroalgae by divers using the quadrat sampling
method. Three quadrats (20 x 20 cm) were randomly scattered in positions of maximum
seagrass coverage (e.g. 100 %). Sediment samples were collected inside vegetated and non-
vegetated sediment by divers using plastic core samplers (15 cm, 15.9 $cm^2$). For
granulometric composition, organic matter, prokaryotic abundance, total lipids and fatty acid
analyses, the cores were cut into 1 cm sections to a depth of 8 cm and lyophilized, except of
sections for prokaryotic abundance analysis, that were weighted (approx. 2 g) and fixed with
formaldehyde (final conc. 4% v/v) immediately after slicing the sediment core.

2.3 Temperature (T) and salinity (S) measurements
T was measured continuously (in 30 min. intervals) using HOBO pendant temp/light Data
Loggers (Onset, USA) which were replaced at each sampling. S was measured on sampling
dates by a pIONneer 65 probe (Radiometer analytical, Copenhagen).

2.4 Inorganic nutrients, Chl *a* and particulate matter (PM) analysis
Nitrate ($NO_3$), nitrite ($NO_2$), ammonia ($NH_4$), phosphate ($PO_4$) and silicate ($SiO_4$) were
analyzed spectrophotometrically according to Strickland and Parsons (1972). Chl *a* was



determined by the fluorometric procedure after filtration of seawater through Whatman GF/F
filters and extraction in 90 % acetone (Holm-Hansen et al., 1965). PM was determined
gravimetrically after filtering up to 5L seawater on pre-weighed, combusted Whatman GF/F
filters which were dried (at 60°C) and reweighed.

2.5 Determining prokaryotic abundance
For determining the prokaryotic abundance in seawater, 2 ml of formaldehyde (final conc. 4%
v/v) fixed samples were stained with 4,6-diamidino-2-phenylindol (DAPI, 1 μg mL$^{-1}$ final
conc.) for 10 min (Porter and Feig, 1980). In sediment samples, prokaryotes were detached
from the sediment particles by addition of Tween 80 (0.05 mL) and ultrasonicated for 15 min
(Epstein and Rossel 1995). After sonication, 1 mL of the supernatant was stained with DAPI
(final conc. 5 μg/mL). DAPI stained samples were filtered onto black polycarbonate filters
(Whatman, Nuclepore, 0.22 μm) and counted under an epifluorescence microscope (Zeiss
Axio Imager Z1).

2.6 Biometry of *C. nodosa* and epiphytic macroalgae
The material from each quadrat was washed under running seawater to remove sediment.
From each quadrat algae, leaves and rhizomes with roots were separated. The length of the
longest leaf on each shoot was measured and the shoots were counted. Species of macroalgae
were determined, and their coverage was estimated according to the Braun-Blanquet scale.
Separated samples were washed with filtered and autoclaved seawater, weighed, dried at 60
°C for 48 h and re-weighed. The dry mass was calculated per area (g m$^{-2}$).

2.7 Granulometric composition of the sediment and its organic matter content
For granulometric analysis of the sediment, each sample was wet sieved through a set of
seven standard ASTM sieves (4-, 2-, 1-, 0.5-, 0.25-, 0.125-, 0.063-mm mesh size). The
fraction that passed through the 0.063-mm sieve was collected and analyzed following the
standard sedigraph procedure (Micromeritics, 2002). The material that was retained on the
sieves was dried and weighted. The data obtained by both techniques were merged to obtain a
continuous grain size range and analyzed with the statistic package Gradistat v 6.0. Sediments
were classified according to Folk (1954). The sediment permeability was calculated based on
median grain size (d$_g$) following the empirical relation by Gangi (1985). The organic matter
content was determined as ignition loss after heating dried sediment sections at 450°C for 4 h
in a muffle furnace.





2.8 Oxygen (O$_2$), hydrogen sulfide (H$_2$S) and redox potential (Eh) profiling
The microprofiles of O$_2$, H$_2$S and Eh were measured on intact cores immediately after
sampling using a motorized micromanipulator (MMS9083) equipped with microsensors OX-
100 and H$_2$S-200, redox microelectrode RD-200 coupled with reference electrode REF-RM
(Unisense A/S, Denmark). Prior to the measurements, the OX-100 microsensor was calibrated
using a two-point oxic – anoxic calibration; H$_2$S-200 was calibrated in fresh Na$_2$S solutions
using eight-point calibration (1µM - 300 µM in a de-oxygenated calibration buffer
(NaAc/HAc, pH <4); RD-200 with REF-RM was calibrated using two point calibration by
simultaneous immersion of electrodes in quinhydrone redox buffers prepared in pH 4 and pH
7 buffers, all according to the manufacturer's recommendation. During measurements,
sediment cores were placed in a pool filled with seawater from the sampling site to maintain
*in situ* temperature. From July to October 2017 H$_2$S was measured spectrophotometrically in
pore waters (Cline, 1969) squeezed out by centrifugation from each section (5 mm) of the
sediment cores.

2.9 Total lipids, fatty acid composition and elemental sulfur (S$^0$)
Lyophilized samples of seagrass tissues, macroalgae, sediment or particulate matter were
weighed and extracted into a solvent mixture of dichloromethane/methanol (DCM: MeOH,
2:1) in an ultrasonic bath at 35°C with three solvent mixture changes. The extracts were
pooled and separated into layers by addition of 0.9% NaCl solution. Lower DCM layers
(containing lipids) were released over Na$_2$SO$_4$ anhydride, collected in pre-weighed round
bottom flasks and evaporated to dryness using rotavapor. After evaporation, flasks were re-
weighed, and total lipid concentrations (TL, mg g$^{-1}$ DW) were calculated from the difference
in weight. For fatty acids determination, lipid extracts were saponified (1.2 M NaOH in
methanol), acidified (6 M HCl), methylated (14% BF$_3$ in methanol) and extracted into DCM.
Fatty acid methyl esters (FAME) were analyzed by Agilent gas–liquid chromatography
(GLC) 6890 N GC System equipped with a 5973 Network Mass Selective Detector, capillary
column (30 m x 0.3 mm x 0.25 µm; cross-linked 5 % phenylmethylsiloxane) and ultra-high
purity helium as the carrier gas. The GLC settings were as follows: programmed column
temperature rise from 145°C by 4°C/min to 215°C, then by 1°C/min to 225°C and finally by
4°C/min to 270°C at constant column pressure of 2.17 kPa. Retention times, peak areas and
mass spectra were recorded on the ChemStation Software. FAME were identified by mass
spectral data and family plots of an equivalent chain length (ECL) for GC standards. Applied
GC standards were: FAME mix C18–C20, PUFA1, PUFA3 standards (Supelco/Sigma-



Aldrich, Bellefonte, PA, USA); C4–C24 FAME standard mix, cod liver oil and various
individual pure standards (Sigma, Neustadt, Germany).

The following indices of fatty acid profiles were calculated: saturated fatty acids (SAT),

monounsaturated fatty acids (MUFA), polyunsaturated fatty acids (PUFA) and the
unsaturation degree (UND). UND was employed to evaluate the degree of organic matter
degradation due to more susceptibility of unsaturated, particularly polyunsaturated,
components to degradation and calculated according to the formula
[1*(% mono-)+2*(% di-)+3* (% tri-)+4*(% tetra-)+5*(% penta-)+6*(% hexa-enoic)]/% SAT
(Pirini et al., 2007). To evaluate the input of terrestrial organic matter relative to that of
marine origin in particulate matter, the terrestrial to aquatic acid ratio (TAR= C24+C26+C28 /
C12+C14+C16) was used (Cranwell et al., 1987; Bourbonniere and Meyers, 1996).

In FAME chromatograms elemental sulfur ($S^0$), eluted as $S_8$ (m/z 256), was identified by

comparison of retention time and characteristic fragment ions in samples and standard
solutions. The concentration of $S^0$ was estimated on the base of the calibration curve prepared
for standard solution of $S_8$ (Aldrich, Germany) in cyclohexane (2-20 mg $L^{-1}$). The calibration
curve was determined under the same GLC settings as FAME. Limit of detection (LoD) and
limit of quantitation (LoQ) were calculated from the parameters of the calibration curve
constructed on the basis of the 3 lowest concentrations in 3 replicates. LoD and LoQ (0.92 mg
$L^{-1}$ and 2.80 mg $L^{-1}$, respectively) were more than twice the values obtained by Rogowska et
al. (2016) probably due to higher injector and column temperature used in this study than they
proposed as optimal for S determination.

2.10 Data analyses
A multivariate analysis, hierarchical clustering and K-means methods (Systat 12) was applied
to group *C. nodosa* above- and below-ground tissues according to the similarity of their fatty
acid profiles and indices, i.e., physiological condition during the investigated period.

Sediment data were analyzed for two groups of sediment layers, the upper layer (0- 4 cm)

where most of rhizomes and roots are located, and the lower layers (5-7 cm). Differences
between vegetated and non-vegetated sediment samples in each sediment layer were tested by
one-way ANOVA. Correlations among parameters were tested using the Pearson's correlation
coefficient (r). The level of statistical significance was $p < 0.05$. A multivariate principal
component analysis (PCA, Primer 6) was applied to identify the most important variables
explaining differences between vegetated and non-vegetated sediments. Correlation matrices
were constructed using variables: $H_2S$, Eh, $O_2$, $S^0$, PA, TL and UND. All variables were



normalized due to their different scales. Only the principal components with eigenvalues >1
were considered.

**3   Results**

3.1 Water column
3.1.1 Environmental variables
During summer of 2017 daily means of sea-bottom temperature in *C. nodosa* meadow ranged
between 26°C and 28°C. During autumn seawater temperatures decreased below 12°C until
the end of December. The coldest period was recorded at the beginning of March lasting only
for a few days (min. 8.62°C). From April to mid-July 2018, temperature increased with
moderate fluctuations to the maximum of 29.26°C recorded in August 2018 (Fig. 1a).

Concentrations of inorganic nutrients and Chl *a* were generally low. The highest

concentrations (DIN: 8.27 µM; $PO_4$: 0.18 µM; $SiO_4$: 9.82 µM; Chl *a*: 0.89 µg $L^{-1}$) associated
with the lowest salinity (34.2) were found in September 2017 (Table S1). The abundance of
prokaryotes (2.6-11.3 x $10^5$ cell $mL^{-1}$) varied seasonally and significantly correlated to
seawater temperatures (r = 0.618; p < 0.05). In contrast, salinity (S: 34.2 - 38.5) and
concentrations of particulate matter (PM: 3.84 - 14.21 mg $L^{-1}$) showed irregular variations
(Fig. 1b) and a significant opposite trend (r = -0.630; p < 0.05).

The particulate lipids exhibited the highest unsaturation degree (UND) during

summer/early autumn 2017 and small increases of UND in April and September/October
2018 (Fig. 1c). UND was significantly correlated with Chl *a* (r = 0.603; p < 0.05). In contrast,
terrestrial to aquatic ratio (TAR) considerably increased in April and was the highest in
August 2018 (Fig. 1c). TAR was negatively correlated to UND (r = -0.644, p < 0.05) and
positively to particulate matter (r = 0.641, p < 0.05). Although PUFA with 18 C atoms made
the largest contribution to the total PUFA pool, C20 PUFA, mainly of phytoplankton origin,
showed a similar trend as observed for UND (Fig. S1, Table S2).

3.2 *Cymodocea nodosa* meadow
3.2.1 Biometry
*C. nodosa* leaves and shoots reached the highest biomass (285.3 ± 57.4 g $m^{-2}$), length (102.4 ±
26.6 mm) and shoot density (3703±334 shoots $m^{-2}$) in October 2017 (Fig. 2a). After the
appearance of the regular vegetation minimum in November 2017, biometric indices further
decreased reflecting the decay of the meadow in summer 2018. In August 2018, only yellow
to brownish leaves on sparse shoots were collected (4.5 ± 1.3 g $m^{-2}$, 5.4 ± 1.3 mm and 30 ± 35



shoots m$^{-2}$). In September and October 2018, no shoots or leaves were observed (Fig. 2a). The
biomass of rhizomes and roots reached also its maximum in October 2017 (599.7 ± 36.8 g m$^{-}$
$^2$). In contrast to leaves and shoots, the belowground biomass was stable until March 2018
when a decline was observed that continued until October 2018 (30.5 ± 6.8 g m$^{-2}$) (Fig. 2a).

3.2.2 Total lipid (TL) concentrations and fatty acid composition
TL in the *C. nodosa* aboveground tissue (6.7 - 25.3 ± 2.4 mg g$^{-1}$ DW) increased until February
2018, when maximum TL concentrations were measured (Fig. 2b). Thereafter, TL
concentrations decreased until August 2018. During this period, the belowground TL
concentration (6.3 ± 1.9 – 15.9 ± 1.1 mg g$^{-1}$ DW) was generally lower than the aboveground
TL concentrations and the trend was similar to that of leaves. The minimum concentrations of
TL were observed in September 2018, while in October 2018, concentrations similar to that
measured in October 2017 were observed (Fig. 2b).

The major fatty acid components in *C. nodosa* tissues were palmitic (C16:0) amongst the

saturated (SAT) and oleic (C18:1n-9) in monounsaturated fatty acids (MUFA). In the
aboveground tissue, the main polyunsaturated fatty acids (PUFA) were α-linolenic (C18:3 n-
3, ALA) and linoleic (C18:2 n-6, LA), while in the belowground tissue LA was dominant
(Fig. 2b). The dynamics of UND in the aboveground tissue was principally influenced by
changes in ALA and LA. LA/ALA ratios were < 1 from July 2017 to March 2018, and > 1
from April to July 2018 (Fig. 2b). In August 2018, the LA/ALA ratio was infinite due to the
absence of ALA (Fig. 2b). Elemental sulfur (S$^0$) was detected only in decaying leaves in
August 2018 (0.21 mg g$^{-1}$ DW). In the belowground tissue, S$^0$ was detected in all samples
(Fig. 2b). Higher concentrations were measured during summer 2017 (up to 0.39 ± 0.06 mg g$^{-}$
$^1$ DW). S$^0$ increased from minimum concentrations in April (0.02 ± 0.01 mg g$^{-1}$ DW) until
September 2018 reaching 1.42 mg g$^{-1}$ DW (Fig. 2b).

According to the fatty acid profiles, *C. nodosa* leaves were classified in three groups,

except for the leaves collected in August 2018 (Fig. 3). The most distinguishing features
specifying physiological differences between Group 1 (July - October 2017 and February -
March 2018), Group 2 (November - December 2017 and April - May 2018) and Group 3
(June and July 2018) were decreasing mean values of PUFA, UND, ALA and LA and
increasing means of SAT and the proportion of long-chain saturated fatty acids (C ≥ 24). In
the ungrouped leaves from August 2018 ALA was not found, PUFA and UND were at a
minimum, while SAT and C ≥ 24 at a maximum (Table S3). Three groups of rhizomes and
roots (Group 1: July - October 2017 and February - March 2018; Group 2: November -



December 2017 and April - May 2018 and Group 3: (June - October 2018) showed similar
characteristics to the groups 1, 2 and 3 of related leaves (Table S4).

3.2.3 Epiphytic macroalgae
From July 2017 to February 2018 different taxa of macroalgae belonging to the three phyla
Chlorophyta (*Halimeda tuna*, *Dasycladus vermicularis*, *Cladophora prolifera*, *Udotea*
*petiolata*), Rhodophyta (*Rytiphlaea tinctoria*, *Peyssonnelia* spp, *Gelidium* sp.) and
Ochrophyta (*Dictyota dichotoma*) were covering the meadow in varying proportions and
abundances (Fig. 4). After March 2018, when only few individuals of *Peyssonnelia* sp. were
found, macroalgae were no longer present in the *C. nodosa* meadow.
Although the fatty acid profiles of macroalgal communities were highly variable, the
contribution of 18- and 20 PUFA to the total PUFA pool generally depended on the prevailing
phyla and their characteristic PUFA pattern. The algae belonging to Rhodophyta and
Ochrophyta are richer in 20 PUFA (C20:5n-3, C20:4n-6), while Chlorophyta are generally
showing prevalence of 18 PUFA (C18:3n-3, C18:2n-6) (Schmid et al., 2014). Furthermore,
their contribution to biomass varied due to large differences in morphology, which most likely
also contributed to the variability of fatty acid profiles. 18 PUFA and 20 PUFA showed the
highest contribution to the total PUFA pool during the dominance of Chlorophyta and
Rhodophyta in the macroalgal community, respectively. In most samples, the lowest
contribution to the total PUFA pool was observed for 16 PUFA and 22 PUFA (Fig. S2).

3.3 Sediment
3.3.1 Granulometric composition
According to the granulometric composition, median grain sizes ($d_g$) and permeability (k) the
vegetated and non-vegetated sediments were classified as slightly gravelly sandy mud (g)sM,
fine grained ($d_g < 165$ μm) and low permeable to impermeable sediment ($k < 2 \cdot 10^{-11}$ m$^2$). In
general, the *C. nodosa* sediment consisted of a significantly higher proportion of sand (Sa),
and lower proportion of silt (Si) and clay (C) (Sa, 41.11 ± 4.34 %; Si, 46.44 ± 2.86 %; C, 9.63
± 2.76 %) in comparison to non-vegetated sediment (Sa, 20.53 ± 10.49 %; Si, 53.24 ± 6.76 %;
C, 23.29 ± 4.86 %). The median grain size and permeability in *C. nodosa* sediment ($d_g$, 37.51
± 17.97 μm, k, $1.22 \cdot 10^{-12} \pm 1.13 \cdot 10^{-12}$ m$^2$) were significantly higher than in non-vegetated
sediment ($d_g$, 10.86 ± 5.34 μm; k, $1.04 \cdot 10^{-13} \pm 1.02 \cdot 10^{-13}$ m$^2$). The upper layers of both cores
(0 - 4 cm) had larger particles, while the lower layers (5 - 8 cm) showed a uniform distribution
of smaller grain sizes (Fig. 5).



3.3.2 $O_2$, $E_h$, $H_2S$ and $S^0$
Oxygen concentrations ($O_2$) in the bottom water of the *C. nodosa* meadow varied in a wide
range (0 µM - 171.4 ± 17.6 µM) and generally followed the $O_2$ saturation trend (Fig. 6a).
From May to June 2018, $O_2$ decreased below 62.5 µM, considered as severe hypoxia (Vaquer-
Sunyer and Duarte 2008) and was completely depleted in July 2018 (Fig. 6a). From August to
October 2018, $O_2$ increased again. The variations of $O_2$ in the bottom water of the non-
vegetated sediment were similar to those in the *C. nodosa* meadow albeit generally higher
(79.4 ± 10.4 µM – 212.2 ± 33.4 µM) than in the vegetated sediment except for September and
October 2018 (Fig. 6a).

In general, $O_2$ penetration depth in the vegetated and non-vegetated sediment co-varied

with the $O_2$ concentration in the bottom layer, penetrating deeper when its concentration in the
bottom water was higher (Fig. 6b). In the vegetated sediment, $O_2$ was mainly depleted down
to 1 cm of depth. In the non-vegetated sediment, the oxygen penetration depth was up to 4
times higher than in vegetated sediments, except for the period from August 2018 to October
2018 when the penetration depths were similar (Fig. 6b).

The thickness of the oxic (Eh > 150 mV) and suboxic (150 mV > Eh > 0 mV) layers in the

vegetated sediment increased from July 2017 (~ 0.5 cm) to March 2018 (~ 4 cm), and
decreased progressively from April (~ 0.8 cm) towards the surface in July 2018, when the
entire sediment core was anoxic (Eh < 0). From August (~ 1 cm) to October 2018 (~ 2.5 cm)
the oxic and suboxic layer thickness increased again (Fig. 7). Oxic conditions (Eh > 0)
generally reflected $O_2$ concentrations in the bottom waters. The dynamics of Eh in non-
vegetated sediment were similar to those in the vegetated sediment. However, the thickness of
the oxic layer was considerably larger than in the vegetated sediment. Reducing conditions
(Eh < 0) were only recorded in July and August 2017 (Fig. 7).

Concentrations of free $H_2S$ in the pore water of the vegetated sediment generally increased

with depth creating an accumulation zone mainly within the upper sediment layers (1 - 4 cm)
(Fig. 7). From July to November 2017, $H_2S$ concentrations increased up to 120 µM (at 4 - 5
cm). In December 2017, $H_2S$ was low and uniformly distributed throughout the core (< 5
µM). $H_2S$ concentrations increased and the accumulation layer was ascending from March (up
to 34.2 ± 12.8 µM; 5 - 7 cm) to April 2018 (up to 177.2 ± 125.1 µM; 3.5 - 4.5 cm). During
May 2018 (up to 107.8 ± 75.9 µM; 2.5 - 4 cm), June (up to199.0 ± 6.3 µM; 1.5 - 6 cm) and
July (up to 210.1 ± 138.9 µM; bottom water - 6 cm) a propagation of the accumulation zone
was observed in addition to an increase in $H_2S$ (Fig. 7). In August 2018 (up to 1164.1 ± 702.1
µM; bottom water - 7 cm) extremely high concentrations over the entire sediment core were



recorded. In September and October 2018, $H_2S$ concentrations decreased (down to $140.0 \pm$
$25.3$ and $72.7 \pm 52.7$ µM; bottom water - 7 cm and 1 - 7 cm, respectively). In the non-
vegetated sediment, $H_2S$ depth profiles were similar to those in vegetated sediments, but the
concentrations were generally lower, except for the summer of 2017 when the concentrations
were comparable but the accumulation zones deeper (Fig. 7).
$S^0$ mainly occurred in oxic (Eh > 150 mV) and suboxic (150 mV > Eh > 0 mV) layers of
both, vegetated and non-vegetated sediments (Fig. 7). Generally, the ranges of approximated
$S^0$ concentrations in vegetated sediment ($8.5 \cdot 10^{-5}$ - $0.39$ mg·g$^{-1}$ DW ~ $2.6 \cdot 10^{-3}$ - $12.1$ µmol·g$^{-1}$
DW), except for the extreme value in April 2018 ($0.99$ mg·g$^{-1}$ DW ~ $30.8$ µmol·g$^{-1}$ DW),
were similar to those found at the non-vegetated sites ($2.9 \cdot 10^{-4}$ – $0.28$ mg·g$^{-1}$ DW ~ $9.2 \cdot 10^{-3}$ –
$8.9$ µmol·g$^{-1}$ DW).

3.3.3 Prokaryotic abundance
Prokaryotic abundance varied largely in vegetated ($2.1$ - $39.9 \cdot 10^7$ cells g$^{-1}$ fresh weight, FW)
and non-vegetated sediments ($3.7$ - $24.1 \cdot 10^7$ cells g$^{-1}$ FW). Prokaryotic abundance was
significantly higher in the upper than the lower layers of vegetated (F = 40.553, p < 0.05) and
non-vegetated (F = 52.531, p < 0.05) sediments (Fig. 8). Prokaryotic abundance showed
significant monthly changes in the upper (F = 3.053, p < 0.05) and lower layer (F = 5.035, p <
0.05) of vegetated sediments, in contrast to both layers of non-vegetated sediments (p > 0.05).
Prokaryotic abundances were significantly higher in the upper layers (F = 44.577, p < 0.05)
and significantly lower in the lower layers (F = 5.986, p < 0.05) of vegetated than in the
respective layers of non-vegetated sediments (Fig. 8). In the upper sediment layer, prokaryotic
abundances were significantly higher in the vegetated than in the non-vegetated sediments
from July to October 2017 and from June to August 2018 (Fig. 8). In the lower layers of
vegetated sediments, prokaryotic abundance was significantly higher than in the non-
vegetated sediments in October 2017 and in August and September 2018 (Fig. 8).

3.3.4 Organic matter, total lipids and fatty acid composition
The concentrations of organic matter (OM) and total lipids (TL) were highly correlated in
vegetated (OM: 37.6 - 231.1 mg/g DW, TL: 0.15 - 2.75 mg/g DW; F = 214.172, p < 0.05) as
well as in non-vegetated sediments (OM: 56.7 - 160.3 mg/g DW, TL: 0.33 - 2.39 mg/g DW; F
= 45.569, p < 0.05). OM and TL generally decreased with depth and exhibited similar
changes throughout the investigated period with significantly higher concentrations in upper
than in lower sediment layers (p < 0.05) (Fig. 9).



In the vegetated sediment, TL showed significant monthly changes in the upper (F =
11.418, p < 0.05) and lower sediment layers (F = 3.186, p < 0.05), in contrast to both layers of
non-vegetated sediment (p > 0.05). From July to October 2017, in the upper layer of vegetated
sediments, TL was significantly higher than in non-vegetated sediments (Fig. 9). From
November 2017 onwards, TL decreased slightly until April 2018, reaching similar
concentrations as TL in non-vegetated sediments (Fig. 9). TL concentrations decreased
markedly in May and continued until August 2018. During that period, TL in vegetated
sediments was significantly lower than in non-vegetated sediments. In September and October
2018, TL concentrations in vegetated sediments were similar to those in non-vegetated
sediment (Fig. 9).
The fatty acid composition of vegetated and non-vegetated sediments was similar and in
both layers characterized by the prevalence of SAT (vegetated upper: 71.2 - 90.4%, lower:
75.9-89.1%; non-vegetated upper: 71.2-80.7%, lower: 78.2-82.5%) over MUFA (vegetated
upper: 7.6-22.9%, lower: 9.0-19.9%; non-vegetated upper: 17.8-24.1%, lower: 15.3-18.2%)
and PUFA (vegetated upper: 1.9-6.9%, lower: 1.9-5.1%; non-vegetated upper: 1.7-4.8%,
lower: 1.7-3.9%). The trends of the monthly changes in UND were similar in both layers of
both sediment types. Those variations were less pronounced in the non-vegetated sediment
where UND varied in narrower ranges in both layers (upper: 0.26-0.51, lower: 0.23-0.33) than
in vegetated sediment (upper: 0.13-0.57, lower: 0.14-0.37). From July to October 2017 and in
April 2018, UND was higher in the upper layers of vegetated sediment than in non-vegetated
one, while from November 2017 to March 2018, UNDs of both sediments were lower than in
previous period (Fig. 9). From June to August 2018, UND decreased considerably in
vegetated sediment, being lower than in non-vegetated sediments. During September and
October 2018, an increase of UND was observed in both sediments. In the lower layers,
UNDs were similar, except for July and August 2018 when a considerable decrease of UND
was observed in vegetated sediments (Fig. 9).
The proportions of PUFAs with chain lengths of 16, 18, 20, and 22 C atoms within the
PUFA pool were similar between the respective layers of both sediments. Throughout the
study period, the highest contribution of 18PUFA originated from *C. nodosa* detritus and
Chlorophyta was observed (Fig. S3, Table S2). From July to October 2017, April to May
2018 and September to October 2018, a contribution of 20PUFA attributed to phytoplankton
and Rhodophyta was also detected. 16PUFA and 22PUFA accounted for the smallest
contribution to the PUFA pool and were found in seston and macroalgae (Fig. S3, Table S2).





The similarities between the sediments were also observed in the contribution of the main
SAT components to the SAT pool from July 2017 to March 2018 and from September to
October 2018 (Fig. S3, Table S2). From April to August 2018, an increase of the long-chain
($C \geq 24$) and common (C16:0 + C18:0) fatty acids followed by the decrease of bacterial fatty
acids (BACT) contribution to the SAT pool was observed in both layers of the vegetated
sediment. In contrast, the contribution of these components to the SAT pool was fairly
invariable in non-vegetated sediments during the same period (Fig. S3, Table S2).

3.3.5 Relationship between different physicochemical parameters
The relationships between $H_2S$, $O_2$, TL, $S^0$, PA, Eh and UND in vegetated and non-vegetated
sediment are shown in the principal component analysis, where PC1 explained 42.5 % and
PC2 14.4 % of variability (Fig. 10). The loadings for positive relationships were obtained for
$H_2S$ (0.298) on PC1 and Eh (0.541) and $O_2$ (0.327) on PC2. For the negative relationships, the
loadings were for TL (-0.534), UND (-0.494), $S^0$ (-0.388), Eh (-0.327), PA (-0.296) and $O_2$ (-
0.191) on PC1, and $H_2S$ (-0.536), $S^0$ (-0.485), TL (-0.165) and UND (-0.221) on PC2.
PC1 separated most of the upper sediment layers (July 2017 - May 2018, September -
October 2018) according to the higher concentrations of TL and $S^0$, higher UND and more
positive Eh from the most of the lower layers and upper layers of vegetated sediments (June -
August 2018) with increased $H_2S$ concentrations. On PC2, the vegetated was separated from
the non-vegetated sediment due to higher concentrations of $H_2S$, $S^0$ and more negative Eh,
which characterized vegetated sediments during almost the entire study period. The extreme
concentrations of $S^0$ and $H_2S$ found in the upper layer in April and the lower layer in August
2018, respectively, were responsible for the considerable separation of these layers from all
other vegetated layers (Fig. 10).

**4  Discussion**
Saline Bay is a shallow, highly dynamic coastal area characterized by frequent turbid waters
due to the combined effect of land run-off and wind-driven resuspension of fine sediment.
Nutrients and Chl *a* (as a proxy for autotrophic biomass) varied in the ranges characteristic for
the oligotrophic coastal waters off Rovinj (Ivančić et al., 2018). The increases in particulate
matter concentration were associated with freshwater input, while their enrichment with
unsaturated fatty acids deriving from phytoplankton was observed during the increases of
autotrophic biomass. However, only in September 2017, this increase was supported by
nutrients from the water column, while all other less pronounced increases were most likely



connected to bottom waters where phytoplankton could have been supplied with nutrients
made available through sediment resuspension. In accordance, increases in the particulate
lipid matter of terrigenous origin have been observed, being generally elevated from April to
August 2018. Therefore, during this investigation the dynamics of the particulate matter was
most likely under the combined influence of terrigenous input and sediment resuspension,
including detritus from the *C. nodosa* meadow.
In temperate Mediterranean coastal waters *C. nodosa* meadows show a clear unimodal
annual growth cycle, reaching maximum development in summer, and minima during winter
and a particularly active growth phase in spring (Terrados and Ross, 1992; Zavodnik et al.,
1998; Agostini et al., 2003). In Saline Bay, the maximum growth was shifted towards early
autumn. This shift was most likely due to the prevalence of massively grazed leaves during
July and August 2017, suggesting an intense grazing activity in the meadows, which probably
decreased during September and October 2017. A minimum growth occurred during late
autumn/winter, as commonly observed. However, during the spring 2018, phenological
parameters continued to decrease in spite of established favorable environmental conditions
for growth, i.e., increase in water temperature, intensity and period of solar radiation. This
decrease continued until the complete extinction of the aboveground tissue in August 2018.
The belowground tissue followed a similar trend, but with less expressed changes. Still, their
recognizable remnants were found after the loss of the aboveground tissues.
During the summer/early autumn 2017 and winter 2018, an adaptation of *C. nodosa* leaves
to the decreasing solar radiation and temperature occurred, respectively. In both periods, an
increase in unsaturation degree (primarily due to ALA increase) in order to increase the
membrane fluidity was observed. From July to October 2017, the temperature of the water
column was still optimal for elongation of the leaves and biomass increase, while the ambient
light intensities were continuously decreasing. An additional reduction of available light
might occur from the self-shading effect due to high canopy biomass, and/or shading due to
epiphytic macroalgae growth and turbidity of the water column. Desaturation of low and
fairly invariable lipids during the most active growth phase suggested an increase in the
membrane fluidity to optimize photosynthetic activity under low light conditions. Such
physiological adaptation as a response to low light availability was found in seagrasses living
along a depth gradient (Beca-Carretero et al., 2019) and macroalgae in contrasting seasons
(Schmid et al., 2014). During the winter, data indicate a progressive trend toward highest total
lipids as well as the proportions of PUFA. Rapid desaturation of increasing lipids could be
attributed primarily to a sharp and continuous decrease in water temperature. An increase in





the level of PUFA is considered to provide a mechanism for the thermo-adaptive regulation of
membrane fluidity and cold resistance in algae and plants (Terrados and Lopezjimenez, 1996;
Iveša et al., 2004; Upchurch, 2008).

In contrast, in late autumn 2017 and spring 2018, the decrease in PUFA and UND

indicated a reduced fluidity and activity of photosynthetically active membranes. The lower
fluidity reduces proton leakage through the thylakoid membranes and energy consumption for
their maintenance (Quigg et al., 2006; Wacker et al., 2016). The reduced photosynthetic
activity was associated with a decreased abundance of shoots and aboveground biomass.
During the period of reduced growth and shedding leaves and shoots the plant further
balances metabolic requirements and mobilize energy from the carbohydrate reserves stored
in the belowground tissue (Alcoverro et al., 2001; Lee et al., 2007). However, major
differences are observed between the two periods indicated by the LA/ALA ratios. During
November and December, LA and ALA proportionally decreased by keeping their ratio < 1,
while during April and May ALA decreased while LA remained stable. The resulting
LA/ALA > 1 suggests a decrease in the conversion of LA to ALA, which occurs in conditions
of light reduction (Harris and James, 1965). This finding apparently contradicts the adaptation
to low light conditions observed during *C. nodosa* healthy and regular growth and suggests
the reduction of light below the minimum requirements for *C. nodosa* survival. Such
conditions of light deprivation existed in April 2018, when the plant had been most probably
exposed to increased siltation, due to a rise in terrigenous input combined with resuspension
of sediment provoking elevated autotrophic growth. The intensive siltation is associated with
the increased light attenuation, both through the direct shading effect of suspended sediments
and through the promotion of phytoplankton and epiphyte growth by the associated increase
in nutrients (Terrados et al., 1998; Halun et al., 2002; Brodersen et al., 2015). Therefore, the
increase in seawater turbidity and considerable sediment re-deposition on the leaves might
have severely impaired the light availability and slowed down the plant's photosynthetic
activity. When the minimum light requirements (~14% of incidence light) are not met, *C.*
*nodosa* intensely sheds leaves and shoots, while at light level of < 1% of surface solar
radiation the plant dies off (Collier et al., 2012). This reduced light condition apparently
persisted until May 2018 and most likely prevented the re-establishment of photosynthesis
and *C. nodosa* continued to shed shoots and leaves.

During June and July, the increase in LA/ALA ratio in the leaves and overall saturation of

decreasing lipids in above- and below-ground tissues indicated a sudden and significant
deterioration of the physiological conditions of *C. nodosa*. Additionally, the loss of leaf tissue





negatively impacted the photosynthetic carbon fixation and therefore oxygen production,
including the transport of oxygen to belowground tissue (Lee and Dunton, 1997; Lee et al.,
2007). The belowground tissue that was not supported by photosynthetically derived oxygen
became anaerobic. The induced anaerobiosis most likely caused a complete inhibition of the
fatty acid desaturation chain (Harris and James, 1965) and a permanent breakdown of
photosynthesis leading to the final decay of the aboveground biomass in August 2018. As a
result, the reduced renewal and storage of energy reserves in the belowground tissue led to a
considerable depletion of reserves and loss of biomass.

In a healthy seagrass meadow, the oxygen generated by seagrass photosynthesis is

transported to belowground tissues to maintain an oxic microsphere around roots and
rhizomes, re-oxidize sulfide to non-toxic $S^0$, thus preventing an invasion of $H_2S$ into the plant
(Pedersen et al., 1998; Holmer et al, 2005). Due to rapid oxygen depletion for respiratory
needs and low storage capacity of lacunae, oxic conditions in belowground tissues are
partially maintained by oxygen diffusing from the water column into belowground tissue
(Pedersen et al., 1998; Greve et al., 2003; Sand-Jensen et al., 2005). An oxic microsphere
around the seagrass roots stimulate the growth of endosymbiotic sulfide-oxidizing
prokaryotes (Jensen et al., 2007), which are regular members of the seagrass microbiome
(Ugarelli et al., 2017; Fahimipour et al., 2017). $S^0$ was found in the *C. nodosa* belowground
tissue during the entire investigation period, as already observed in seagrasses living in
sulfidic sediments (Holmer and Hasler-Sheetal, 2014; Hasler-Sheetal and Holmer, 2015).
However, from July 2017 until March 2018, it seems that the plant was sufficiently supplied
with oxygen produced either by photosynthesis and/or supplied by diffusion from the well-
oxygenated water column. This probably ensured the complete re-oxidation of the potentially
intruding sulfide preventing root anoxia. As photosynthesis and therefore oxygen production
were already reduced in April 2018, the maintenance of the oxic rhizosphere and the internal
$O_2$ partial pressure in the lacunae further depended mainly on the diffusion of $O_2$ from the
water column. From April to June 2018, $O_2$ in the bottom water drastically decreased. Due to
poor supply, $O_2$ content of the belowground tissue was too low to maintain the oxic
microenvironment and therefore, the plant tissues became potentially accessible to sulfide
intrusion (Pedersen et al., 2004). To reach the leaves, sulfide invasion has to exceed
belowground tissue oxidation capacity and pass through these tissues, invading the meristems
located at the base of the leaves, where sulfide toxicity can have drastic effects on shoot
growth and survival (Greve et al., 2003; Frederiksen et al., 2008). In July 2018, the bottom
waters were completely depleted in $O_2$ and the whole plant probably exposed to $H_2S$. $H_2S$



inhibit cytochrome c oxidase by binding to regulatory sites on the enzyme, reducing the rate
of cellular respiration and leading to the chemical asphyxiation (Nichols et al., 2013). In
August 2018, the inflow of freshwaters re-oxygenated the bottom waters enabling $H_2S$
oxidation in leaves, which were, however, already in an advanced stage of decomposition.
During September and October 2018, the penetration of $O_2$ from the water column gradually
led to the recovery of belowground tissue.
In addition to plant activity, sulfide intrusion into seagrasses is controlled by sediment
biogeochemistry and environmental conditions (Frederiksen et al., 2006), while sulfide
concentration in sediments is determined by the rate of sulfate reduction, which in turn
depends on the amount of organic matter and temperature (Moeslund et al., 1994 ). Organic
matter and closely correlated total lipids in the sediment of *C. nodosa* rooted area changed
significantly throughout the investigated period, in contrast to organic matter in non-vegetated
sediment. Nevertheless, considerable but the co-varying unsaturation degree suggests
similarity in the quality and degradation degree of lipid matter at both, the vegetated and the
non-vegetated sites. This covariation indicates an important contribution of detritus imported
from the meadow as a source of organic matter for prokaryotes in non-vegetated sediments.
Close coupling between the seagrass meadow and non-vegetated sites could be expected due
to their proximity and lower organic content of the non-vegetated sediment, which should
enhance the dependence of prokaryotes on the imports of seagrass detritus from the adjacent
meadows (Holmer et al., 2004). Moreover, the non-vegetated sediment in Saline Bay could
readily support the adsorption of imported organic material due to a higher proportion of mud
(silt and clay) and considerably lower median grain size in comparison to the *C. nodosa*
sediment.
*C. nodosa* sediment was significantly enriched with organic matter, characterized by a
higher contribution of unsaturated, more labile components, in comparison to the non-
vegetated sediment layer only during abundant growth of meadow. Also, sestonic material
from the water column is efficiently trapped and accumulates within the meadow (Gacia and
Duarte, 2001), representing an additional source of labile components derived from
macroalgae and *C. nodosa* leaves. Such easily utilizable organic matter, including dissolved
monomeric carbohydrates, leaching out during decomposition of *C. nodosa* leaves stimulates
prokaryotic growth (Peduzzi and Herndl, 1991). This effect could be observed, as prokaryotic
abundance was higher in *C. nodosa* sediments (Fig. 8). In contrast, the lower unsaturation of
lipid matter in the non-vegetated sediment can be explained by its higher instability.





Resuspension and a wider oxic layer could have further suppressed the preservation of
reactive and more labile organic matter in comparison to the *C. nodosa* sediment.
The relatively low accumulation of $H_2S$ (< 30 μM) during the summer and early autumn
2017 indicated that $H_2S$ was apparently rapidly recycled within the rooted area via re-
oxidation by $O_2$ to $S^0$ and/or removal by precipitation with iron compounds. Most of $S^0$ was
found in oxic layers or suboxic/anoxic boundaries, but also anoxic layers in July and October
2017. The oxidation of $H_2S$ could occur spontaneously by chemical reaction with free oxygen
or mediated by sulfide-oxidizing bacteria (Jørgensen, 1977). Usually $S^0$ is the most abundant
sulfide oxidation intermediate, and it accumulates to higher concentrations than other more
reactive compounds (e.g. polysulfide, thiosulfate, tetrathionate, sulfite; Zopfi et al., 2004). In
Saline Bay sediment $S^0$ occurs in ranges typical for sulfidic coastal sediments (Troelsen and
Jørgensen, 1982; Panutrakul et al., 2001; Pjevac et al., 2014). During the active growth of *C.*
*nodosa*, the rhizosphere surrounding sediment was well supplied with photosynthetically
produced oxygen due to radial oxygen leakage. Therefore, in addition to free oxygen available
in pore waters, both, biotic and abiotic re-oxidation of sulfide was most likely supported by
the oxygen supplied via the release from the root to the surrounding sediment (Holmer et al.,
2006). Generally, thermodynamic and kinetic considerations suggest that biological oxidation
far exceeds chemical oxidation of sulfide in most environments (Wasmund et al., 2017).
Moreover, abundant sulfide oxidizing prokaryotes have been detected in marine sediments
surrounding or attaching to seagrass roots (Cucio et al., 2016; Fahimipour et al., 2017).
In November, due to the degradation of organic matter and reduced oxygen production and
leakage in the rooted zone caused by *C. nodosa* senescence, the re-oxidation capacity of the
sediment was greatly decreased. This resulted in considerable accumulation of $H_2S$ (> 100
μM) which extended up to the sediment surface. During winter and early spring, $H_2S$
production generally decreased, likely due to the reduced activity of sulfate reducing
prokaryotes at lower temperatures, and the sediment gradually shifted towards a more
oxidized state. $H_2S$ detected even in within the oxic sediment and in the rooted area in
February 2018 could be attributed to the sediment heterogeneity and the presence of reducing
micro-niches where anaerobic metabolism could occur regardless of surrounding redox
conditions (Jørgensen, 1977; Frederiksen and Glud, 2006). Moreover, it has been found that at
temperatures below 15°C, organic sulfur is more important than sulfate as a sulfide source.
This was explained by a higher temperature coefficient required for sulfate reduction than for
other heterotrophic processes (Jørgensen, 1977).



In April 2018, the sediment was enriched with fresh organic matter derived from increased
autotrophic biomass in bottom waters. In addition to the induction of the bloom, strong
sediment resuspension, most likely by aeration, stimulated the intense oxidation of $H_2S$ that
started to produce in the rooted zone (up to 180 µM, Fig. 7), due to increased activity of
sulfate reducing prokaryotes possibly triggered by the increase in temperature. An increase in
$S^0$ concentration that reached its maximum in the same layer suggests a simultaneous
oxidation of the produced $H_2S$. The sulfide oxidation probably caused oxygen depletion in the
rooted zone and anoxic zone extension up to the sediment subsurface. In May 2018, the
excess of organic matter accumulated in April 2018 was degraded. The concentrations of $S^0$,
detected only in the suboxic layer, considerably decreased possibly by disproportionation or
respiration by members of the sulfate reducing bacteria. $S^0$-disproportionating
*Desulfobulbaceae* and $S^0$-respiring *Desulfuromonadales* are frequently detected in anoxic
coastal sediments (Pjevac et al., 2014).
From June to August 2018, the decomposition of organic matter, encompassing the entire
sediment core, was intensified and accompanied by a large increase in $H_2S$ concentrations (up
to 1200 µM). The degradation process involved rhizomes and roots, as suggested by the
apparent loss of belowground biomass. Such loss typically occurs in the first stage of plant
decay, the leaching phase (Trevathan-Tackett et al., 2017). Readily available, soluble
carbohydrates that largely contribute to the leachate mass (Vichkovitten and Holmer, 2004)
most probably supported the increase in prokaryotic abundance observed in June and July
2018. However, the significant decrease in prokaryotic abundance that coincided with a
maximum degradation of organic matter and $H_2S$ production in August 2018 might indicate
that remaining compounds were not degradable by the sulfate reduction pathway (Arndt et al.,
2013) and needed the presence of prokaryotes specialized in the anaerobic degradation of
refractory compounds, including cellulose and lignin.
During September and October 2018, $H_2S$ concentrations drastically decreased, and the
sediment was gradually enriched in fresh organic matter. Due to the combined effect of
freshened oxygenated water inflow and resuspension which gradually deepened the oxic
layer, re-oxidation of $H_2S$ increased. Biogeochemical studies suggest that most sulfides (80 –
90 %) are eventually re-oxidized, 10 – 20 % are ultimately buried as complexes with iron (i.e.
FeS, $FeS_2$) or with organic matter after sulfurization (Jørgensen, 1977; 1982). $H_2S$ scavenging
with iron and formation of iron sulfides might be more important in Saline Bay, since
terrestrial waters are washing out *terra rossa*, rich in Fe-oxides and oxyhydroxides (Durn,



2003). For this reason, sediment cores were most likely always black with sulfuric odor,
irrespective of $H_2S$ concentrations or presence of vegetation.

**5 Conclusions**
During the regular growth, from July 2017 to March 2018, *C. nodosa* successfully adapted to
the changes of environmental conditions and prevented $H_2S$ accumulation by its re-oxidation,
supplying the sediment with $O_2$ from the water column and/or leaf photosynthesis. Our results
suggest that the *C. nodosa* die-off was most likely triggered in April 2018 by a reduction of
light availability, which severely reduced leaf photosynthesis and the oxidation capability of
belowground tissue. Simultaneously, in the sediment, depletion of oxygen due to intense
oxidation of $H_2S$ occurred, thus creating anoxic conditions in most of the rooted areas. This
synergistic negative effect on the plant performance exposed *C. nodosa* to $H_2S$ intrusion.
During the degradation of dying above- and belowground tissues, which culminated in August
2018, high concentrations of $H_2S$ were produced and accumulated all over the sediment cores,
including bottom waters. An improvement in the oxygen supply in September 2018 led to the
re-establishment of $H_2S$ oxidation and recovery of the belowground tissue.
Even if the sediment conditions improved by the end of the summer 2018, *C. nodosa* has
not been able to recolonize its previously occupied areas in the rest of 2018 and during 2019.
This finding combined with a visible alteration of the water column and sediment is
suggesting a considerable habitat loss. Further research is needed to examine the fate of Saline
Bay meadows remains and an eventual recolonization of the area.

*Author contribution:* Conceptualization: MN, MK and GJH; Investigation: MK, PP, MM, II,
LJI, IF and MN; Formal analysis and Writing - original draft: MN; Writing – review &
editing: MK, GJH, PP, LJI, II, IF and MM.
*Competing interests:* The authors declare that they have no conflict of interest.
*Acknowledgements.* The financial support was provided by the Croatian Science Foundation
to MN (project IP-2016-06-7118, MICRO-SEAGRASS). We sincerely thank J. Jakovčević
and M. Buterer for nutrient and chlorophyll *a* determination, and A. Budiša and I. Haberle for
occasional help during separation and biometry of plant material.



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



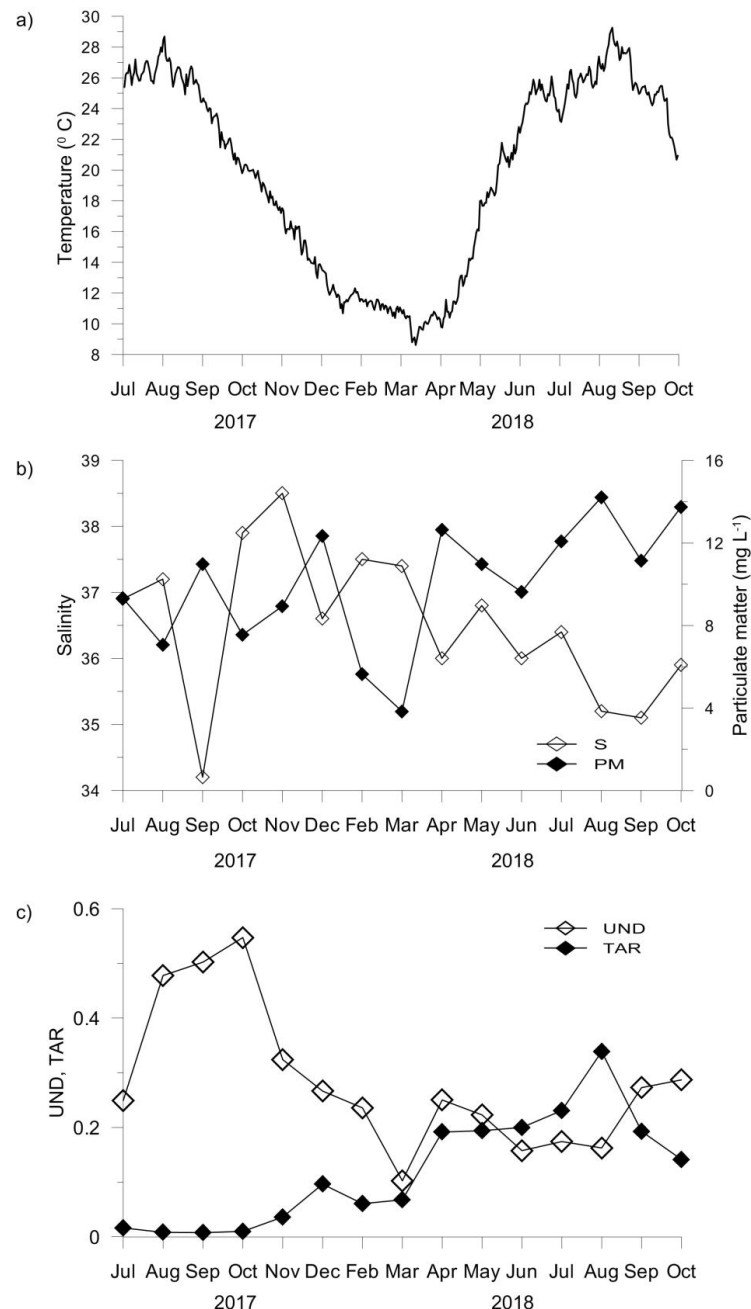

Figure 1. Temperature (a); salinity (b), particulate matter concentration (b); unsaturation degree (UND) and terrestrial to aquatic ratio (TAR) of the particulate lipid matter (c) in seawater.











Figure 2. Above- and below-ground tissue biomasses and shoot density (a), total lipid
concentrations (TL) and linoleic to α-linolenic fatty acids ratios (LA/ALA, an arrow indicates
an infinite value) in above-ground tissue and TL and approximated concentrations of
elemental sulfur ($S^0$) in below-ground tissue (b).





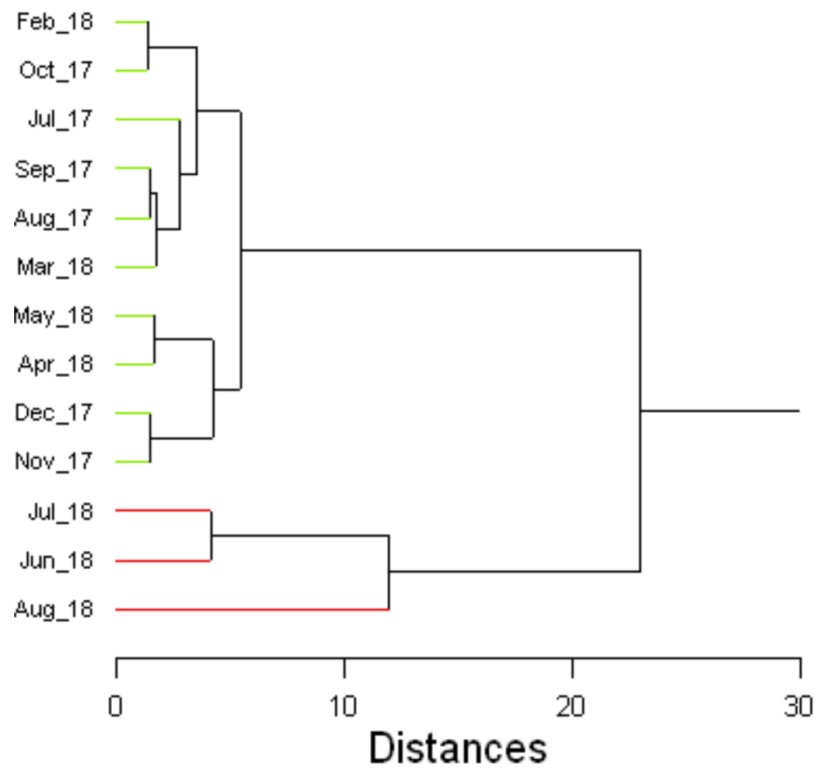

Figure 3. Cluster analysis dendrogram of fatty acid composition of *C. nodosa* leaves.

Summary statistics is given in Table S3.



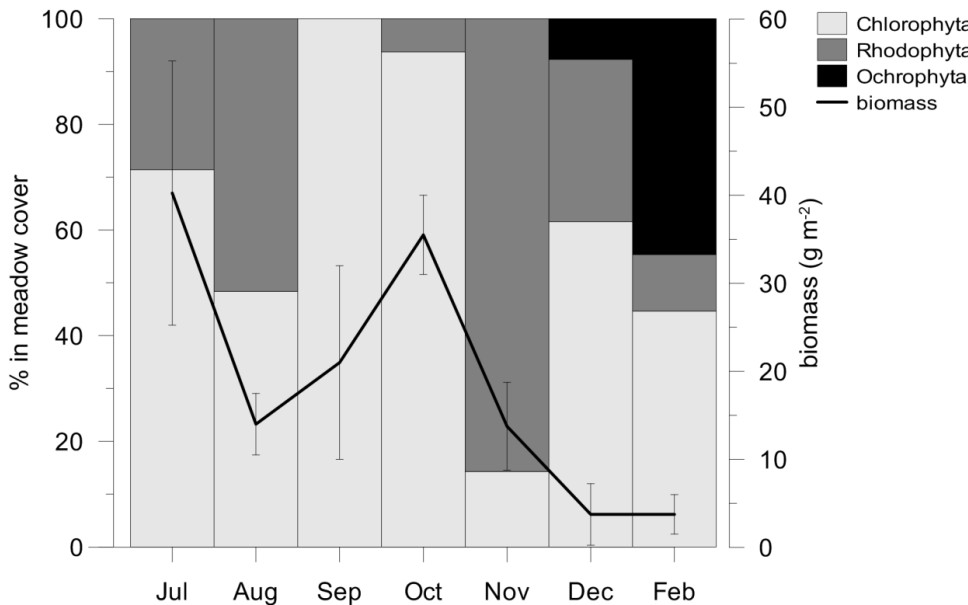


Figure 4. The contribution of macroalgal phyla in a meadow cover and total macroalgal
biomass changes during their notable presence in a *C. nodosa* meadow.

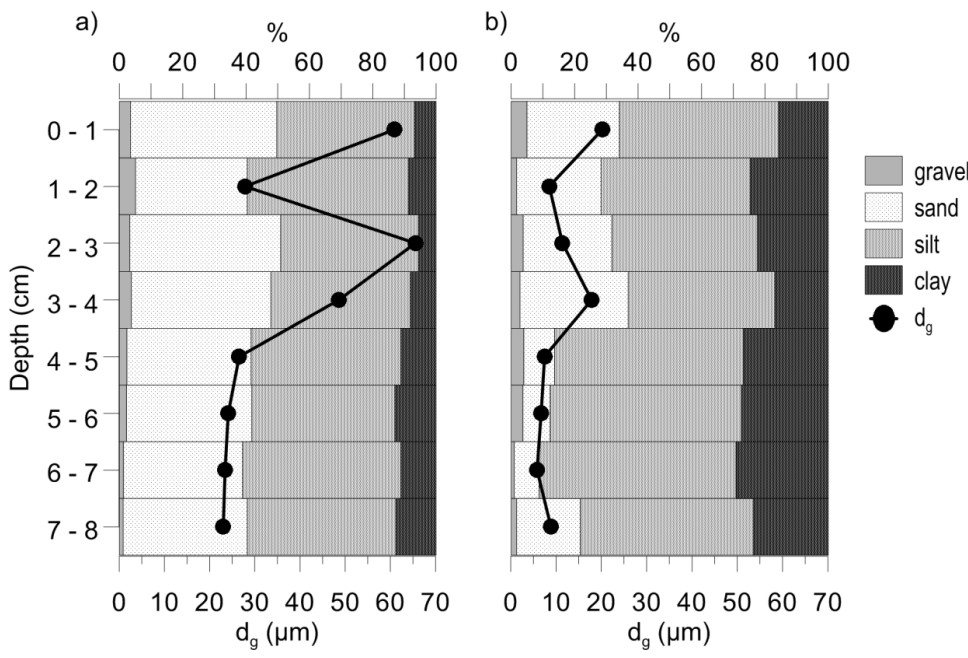


Figure 5. Granulometric composition and median grain size ($d_g$) of vegetated (a) and non-
vegetated sediment (b).





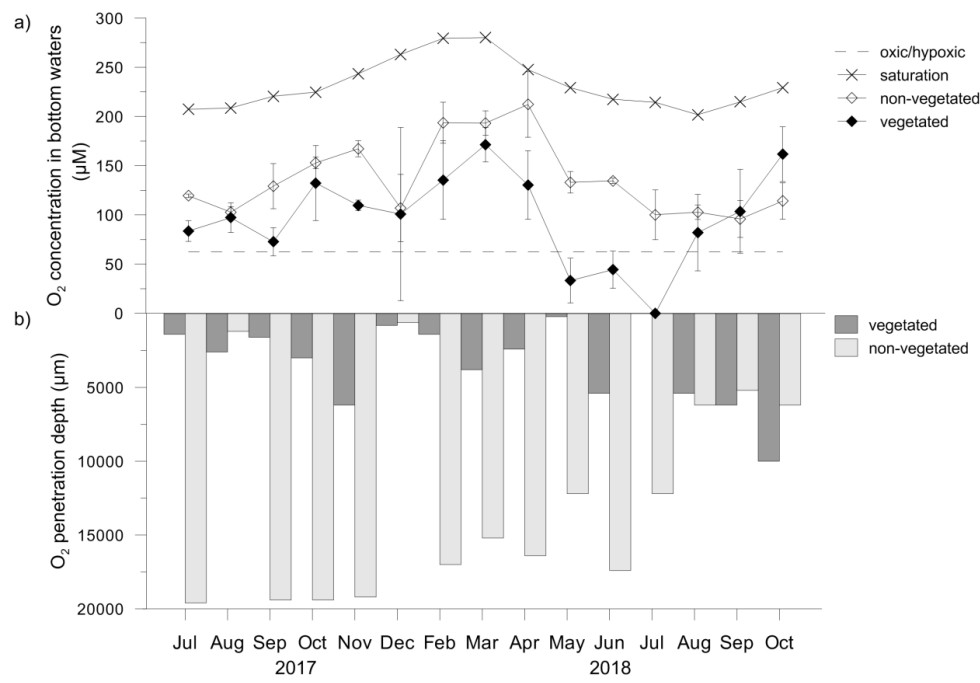

Figure 6. Oxygen concentrations ($O_2$) in bottom waters (a) and $O_2$ penetration depths (b) above and in vegetated and non-vegetated sediment, respectively. $O_2$ at the saturation level was calculated according to the temperature and salinity measured in seawater at the sampling dates; $O_2$ at the hypoxic frontier (~ 62.5 µM) was taken from Vaquer-Sanyer and Duarte (2008).





Figure 7. Depth profiles of H₂S and S⁰ concentrations in vegetated and non-vegetated sediment (adjacent narrow graphs). The redox potential (Eh) in both sediments is shown as areas corresponding to oxic (Eh > 150 mV), suboxic (150 > Eh > 0 mV) and anoxic (Eh < 0 mV) conditions.

962

963

964



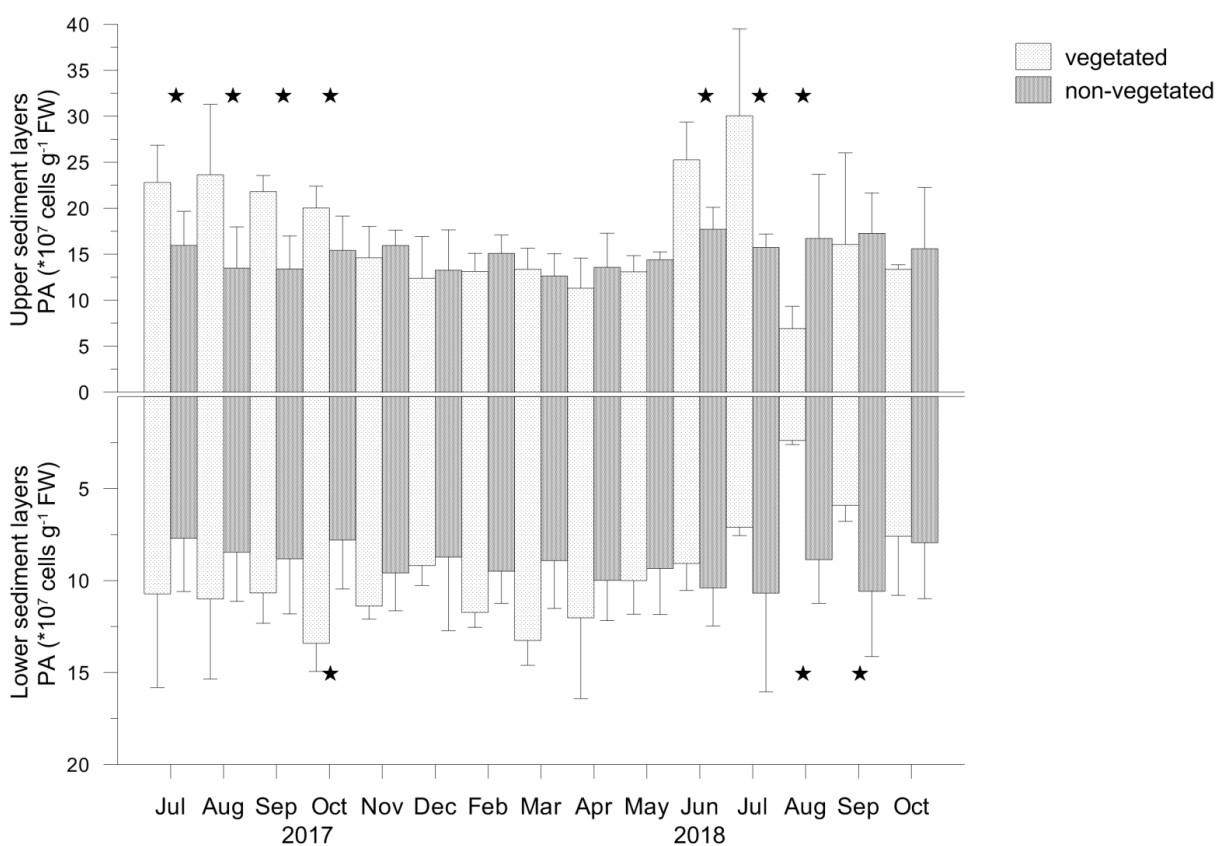

Figure 8. Prokaryotic abundance (PA) in the upper (0 - 4 cm) and lower (5 - 8 cm) layers of vegetated and non-vegetated sediments; significant differences in PA between the sediments are indicated by asterisks.





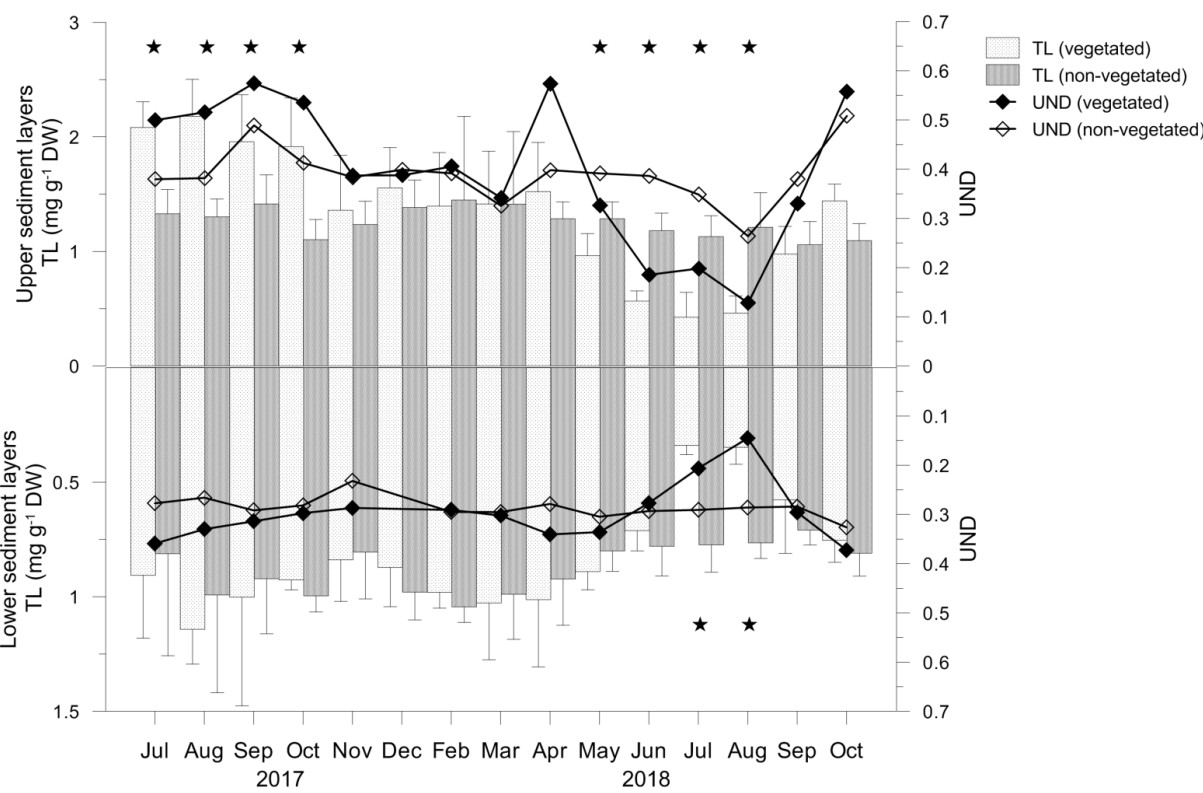

Figure 9. Total lipid concentrations (TL) and unsaturation degree (UND) in the upper (0 - 4 cm) and lower (5 - 8 cm) layers of vegetated and non-vegetated sediments. Significant differences in TL between the sediments are indicated by asterisks.




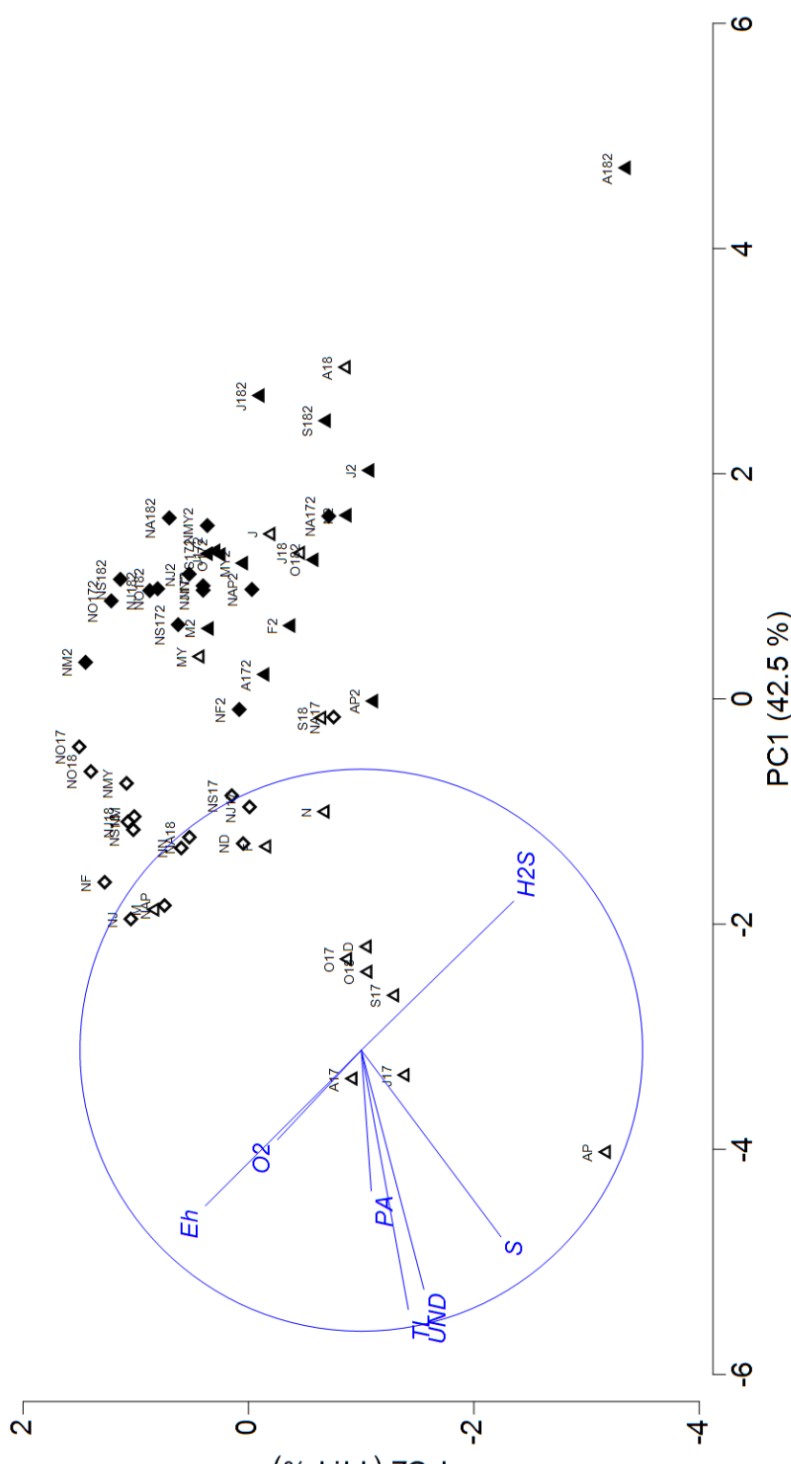


Figure 10. PCA plot of redox potential (Eh), oxygen (O$_2$), hydrogen sulfide (H$_2$S), sulfur (S), total lipids (TL) and prokaryotes (PA)
concentrations and unsaturation degree (UND) in the upper ($0 - 4$ cm; △, ◇) and lower ($5 - 7$ cm; ▲, ◆) layers of vegetated and non-vegetated
sediments, respectively. Projections of variables are given in circles.