# Peer review of "Dynamics of environmental conditions during a decline of a *Cymodocea nodosa* meadow"

_Biogeosciences, 2019_

## Referee Comment (RC1) · Anonymous Referee #1 · 1 Feb 2020

The authors investigated the dynamics of environmental conditions during a decline of a Cymodocea nodosa meadow in the northern Adriatic Sea, analyzed the correlation between those physicochemical and biological parameters, and concluded that the reduced light availability and following photosynthesis was the most likely reason leading to the decay of seagrass meadow. The experiments seem to be conducted carefully and the results were thoroughly discussed. This study supplies helpful information on understanding the decline of seagrass globally. However, there are some points that the authors need to attend to before it can be published in Biogeosciences.

I have two general concerns about this study.

[Figure]

1. The loss of seagrass meadow is attributed to reduced light availability and thus photosynthesis in this study. However, it seems that there is no direct data to support this conclusion. Have the light intensity in water column and photosynthetic rate of seagrasses were measured? The solar radiation in April should not be the lowest level compared to other months.

2. The authors mentioned that from July 2017 to March 2018, C. nodosa successfully adapted to the changes of environmental conditions and prevented H2S accumulation by its re-oxidation, supplying the sediment with O2 from the water column and/or leaf photosynthesis. Then why did not C. nodosa adapt to the environmental changes from April 2018 onwards? I am wondering that the decline of seagrass meadow in the northern Adriatic Sea is a natural process or caused by other drivers?

Specific comments

Line 22 Why did light availability decrease in April 2018?

Line 30 The data in Figure 2 did not show the recovery of the below-ground tissue.

Line 37 Better to supply latest literature as there are loads.

Line 41 Add a comma after matter.

Line 85 A introduction about seagrass meadows in Saline Bay or Adriatic Sea should be supplied here. Meanwhile, it would be helpful to add research gap here.

Line 98 Better to supply a map for the study site.

Line 116 Sampling time/frequency needs to be stated. What is the depth for C. nodosa living?

Line 320 This is true as shown in the green macroalgae Ulva linza (Gao et al. 2018 Food Chemistry, 2018, 258: 71-78).

Line 386 What are these prokaryotic organism?

Lines 485-487 How did you know it? Any data or literature to support this speculation?

Line 680 Conclusion should not be a repetition of Abstract. The purpose of a conclusion is to tie together, or integrate the various issues, findings, arguments etc., covered in the body of the paper, and to make comments upon the meaning of all of it. This includes noting any implications resulting from your discussion of the topic, as well as recommendations, forecasting future trends, and the need for further research.

Line 950 Please annotate which year for the months and explain why it ends in Feb in the legends.

---

## Referee Comment (RC2) · Anonymous Referee #2 · 5 Mar 2020

The paper entitled "dynamics of environmental conditions during a decline of cymodocean nodosa meadow" reported biomass changes of the seagrass along with environmental changes in both seawater and sediments during one year period of 2017 to 2018. The results showed that C. nodosa successfully adapted to the changes of environmental conditions and prevented H2S accumulation by its re-oxidation, supplying the sediment with O2 from the water column and/or leaf photosynthesis, implying that the C. nodosa die-off would be most likely caused in April 2018 by a reduction of light availability. Unfortunately, solar irradiances changes either at the surface of water column or in water were not supplied. Especially, the light levels during investigation periods were not provided. Attenuation of light in water during different seasons

with different amounts of PM can be directly link to photosynthetic performance of the seagrass. While the authors showed that in the sediment, depletion of oxygen due to intense oxidation of $H_2S$ led to anoxic conditions in most of the rooted areas. This could negatively affect respiration of the plant root, therefore, its growth. With reduced growth capacity, high concentrations of $H_2S$ were observed in the sediment cores and bottom waters. This is an interesting result indicating the relationship of $H_2S$ levels with photosynthetic $O_2$ evolution. Generally speaking, the paper has scientific significance, and is suitable to be accepted to BG after revision.

Specific comments: 1. Daily sunlight doses should be in parallel with water temperature, and should be provided, which can be easily obtained from local meteorological stations if the authors did not measure. 2. Decline of the seagrass meadow can hardly be attributed to light availability, grazing pressure or others. It must be a result of multiple drivers impact. Therefore, the discussion should be re-sorted and holistically analyzed. 3. Decreased root respiration may also contribute to the dying off 4. Changes of unsaturated fatty acids could be attributed to many sources, since phytoplankton or microalgae are the main producers of these compounds, it is hard to guess. 5. While grazing rate might be responsible for the changes in seasonal change of stand crop of the seagrass, the authors did not provide any record that grazing rate is higher in the Saline Bay. 6. During summer period, high light and temperature may synergistically reduce the biomass of the seagrass due to higher respiration and higher photoinhibition.

Technical corrections: 1. Repeated wordings should be avoded in a sentence or paragraph. 2. Unit of silicate should be double checked, might be mistaken

Line 83 change shorter to shorten

---

## Author Comment (AC1) · 13 Mar 2020

We greatly appreciate all the reviewers' comments and suggestions. Please find our response letter below. Interactive comment on "Dynamics of environmental conditions during a decline of a Cymodocea nodosa meadow" by Mirjana Najdek et al. Anonymous Referee #1 The authors investigated the dynamics of environmental conditions during a decline of a Cymodocea nodosa meadow in the northern Adriatic Sea, analyzed the correlation between those physicochemical and biological parameters, and concluded that the reduced light availability and following photosynthesis was the most likely reason leading to the decay of seagrass meadow. The experiments seem to be conducted carefully and the results were thoroughly discussed. This study supplies helpful information on understanding the decline of seagrass globally. However, there are some points that the authors need to attend to before it can be published in Biogeosciences. I have two general concerns about this study. 1. The loss of seagrass meadow is attributed to reduced light availability and thus photosynthesis in this study. However, it seems that there is no direct data to support this conclusion. Have the light intensity in water column and photosynthetic rate of seagrasses were measured? The solar radiation in April should not be the lowest level compared to other months. 2. The authors mentioned that from July 2017 to March 2018, C. nodosa successfully adapted to the changes of environmental conditions and prevented H2S accumulation by its re-oxidation, supplying the sediment with O2 from the water column and/or leaf photosynthesis. Then why did not C. nodosa adapt to the environmental changes from April 2018 onwards? I am wondering that the decline of seagrass meadow in the northern Adriatic Sea is a natural process or caused by other drivers?

Authors response: Light intensity in the water column and photosynthetic rates of seagrass were not measured. The reduction of available light to the C. nodosa meadow in April 2018 was indicated by a visible increase in turbidity of the water column (noted in situ by divers), due to an increased concentration of particulate matter of terrigenous origin and sediment resuspension. Terrigenous input in combination with sediment resuspension covered the plant with sediment; thereby significantly reducing the amount of light to the plant below the minimum required for photosynthesis. Besides reduced photosynthesis and therefore supply of the below-ground tissue with oxygen, a depletion of oxygen due to intense oxidation of H2S occurred in the sediment, thus creating anoxic conditions in most of the rooted areas. Most likely, this change in April 2018 drastically and irreversibly compromised the survival of the meadow. From April 2018 onwards, C. nodosa didn't reestablish photosynthesis and continued to lose shoots and overall biomass, while at the same time in the sediment, the concentration of H2S increased considerably, which, due to lack of oxygen, penetrated the plant and caused

the meadow die-off. We would exclude that the process in Saline Bay was natural because several geographically nearby meadows didn't display a similar pattern of regression.

Specific comments: Line 22 Why did light availability decrease in April 2018? Authors response: This sentence in the Abstract was rewritten and now reads: The C. nodosa decline was most likely triggered in April 2018 when light availability to the plant was drastically reduced. Such conditions resulted from increased seawater turbidity due to terrigenous input, sediment resuspension and elevated autotrophic biomass.

Line 30 The data in Figure 2 did not show the recovery of the below-ground tissue. Authors response: We agree with this comment. The sentence in the Abstract now reads: The influx of oxygenated waters in September 2018 led to the re-establishment of H2S oxidation in the sediment and the remaining of the below-ground tissue.

Line 37 Better to supply latest literature as there are loads. Authors response: The following literature has been added. . .e.g. Duarte et al., 2013, Samper-Villarreal et al., 2016

Line 41 Add a comma after matter. . . a comma was added

Line 85 A introduction about seagrass meadows in Saline Bay or Adriatic Sea should be supplied here. Meanwhile, it would be helpful to add research gap here. Authors response: Introduction about seagrass meadows in Adriatic and research gap was added. This paragraph now reads: The seagrass Cymodocea nodosa (Ucria) Ascherson is widely distributed and common species throughout the Mediterranean (Terrados and Ros 1992; Pedersen et al., 1997; Cancemi et al., 2002; Agostini et al., 2003). For the northern Adriatic, however, only sparse data are available on the standing crop, seasonal dynamics or natural/anthropogenic pressures supporting the ecological or conservation status of C. nodosa meadows (Zavodnik et al., 1998; Orlando-Bonaca et al., 2015; 2016). Although C. nodosa show large phenotypic plasticity adapting to diverse natural and anthropogenic stressors by physiological and morphological adaptations, a severe decline has been reported during the last decades in coastal areas (Orth et al., 2006; Short et al., 2011; Tuya et al., 2002; 2014), including the northern Adriatic (Orlando-Bonaca et al., 2015; 2019). One of these declines is documented in our study performed in Saline Bay (northern Adriatic Sea) from July 2017 to October 2018.

Line 98 Better to supply a map for the study site. A map is now provided as Fig S1.

Line 116 Sampling time/frequency needs to be stated. What is the depth for C. nodosa living? Authors response. The sampling dates and depths of C. nodosa were included and now reads: The sampling was performed for 15 months from July 2017 to October 2018. C. nodosa (3 - 4 m of depth) was collected together with rhizomes, roots and macroalgae by divers using the quadrat sampling method.

Line 320 This is true as shown in the green macroalgae Ulva linza (Gao et al. 2018 Food Chemistry, 2018, 258: 71-78). Authors response: The suggested reference was added in the text and listed in References

Line 386 What are these prokaryotic organism? Authors response: Prokaryotic organisms are Bacteria and Archaea from $0.2 - 2\ \mu m$, stained with DAPI and counted by epifluorescence microscopy.

Lines 485-487 How did you know it? Any data or literature to support this speculation? Authors response: In July and August 2017 the leaves were short with visible signs of bites, so we assumed that significantly lower biomass in these two months was the result of grazing activity of herbivores. Generally, the meadows have been shown to be an important source of food for herbivores. We have added the references that support biomass loss by herbivory in C. nodosa meadows; Cebrian et al., 1996; Valentine and Duffy, 2006.

Line 680 Conclusion should not be a repetition of Abstract. The purpose of a conclusion is to tie together, or integrate the various issues, findings, arguments etc., covered

in the body of the paper, and to make comments upon the meaning of all of it. This includes noting any implications resulting from your discussion of the topic, as well as recommendations, forecasting future trends, and the need for further research. Authors response: The Conclusion has been rewritten and now reads: Our results provide insights into the interaction of multiple stressors that have led to the meadow decay, triggered in the sensitive recruitment phase of meadow growth. Even after the improvement of the sediment conditions by the end of the summer 2018, C. nodosa was not able to recolonize its previously occupied areas. This finding combined with a visible alteration of the water column and sediment indicates a considerable loss of the C. nodosa habitat. Further research is needed to examine the fate of Saline Bay meadows and an eventual recolonization of the area. Beyond seagrass itself, this loss had extensive consequences as it has endangered many species that depend on seagrass for food, shelter and nursery. Given the lack of data on the ecological and conservation status of the still numerous seagrass meadows along the northern Adriatic coast, the identification and monitoring of the main pressures acting on them are needed to protect such valuable habitats from degradation and extinction.

Line 950 Please annotate which year for the months and explain why it ends in Feb in the legends. Authors response: The years are annotated and an explanation is added in the Figure 4 legend which now reads: Figure 4. The contribution of macroalgal phyla in a meadow and total macroalgal biomass; after February 2018 macroalgae were no longer present in the C. nodosa meadow.

---

## Author Comment (AC2) · 13 Mar 2020

We greatly appreciate all the reviewers' comments and suggestions. Please find our response letter below. Interactive comment on "Dynamics of environmental conditions during a decline of a Cymodocea nodosa meadow" by Mirjana Najdek et al. Anonymous Referee #2 The paper entitled "dynamics of environmental conditions during a decline of cymodocean nodosa meadow" reported biomass changes of the seagrass along with environmental changes in both seawater and sediments during one year period of 2017 to 2018. The results showed that C. nodosa successfully adapted to the changes of environmental conditions and prevented

H2S accumulation by its re-oxidation, supplying the sediment with O2 from the water column and/or leaf photosynthesis, implying that the C. nodosa die-off would be most likely caused in April 2018 by a reduction of light availability. Unfortunately, solar irradiances changes either at the surface of water column or in water were not supplied. Especially, the light levels during investigation periods were not provided. Attenuation of light in water during different seasons with different amounts of PM can be directly link to photosynthetic performance of the seagrass. While the authors showed that in the sediment, depletion of oxygen due to intense oxidation of H2S led to anoxic conditions in most of the rooted areas. This could negatively affect respiration of the plant root, therefore, its growth. With reduced growth capacity, high concentrations of H2S were observed in the sediment cores and bottom waters. This is an interesting result indicating the relationship of H2S levels with photosynthetic O2 evolution. Generally speaking, the paper has scientific significance, and is suitable to be accepted to BG after revision. Specific comments: 1. Daily sunlight doses should be in parallel with water temperature, and should be provided, which can be easily obtained from local meteorological stations if the authors did not measure. Authors response: Light intensity in the water column was not measured. The temperature of natural water bodies varies in response to diurnal and seasonal changes in solar radiation. Our temperature data showed no unusual diurnal and/or seasonal pattern. Therefore, it could be reasonably assumed that during the study period there was no big difference in the amount of solar radiation received by the sea. On the other hand, the penetration of light in the water column, which regulates photosynthesis, is strongly influenced by the transparency of the water. The attenuation of light in the water column and therefore, the reduced availability of light required for C. nodosa photosynthesis in April 2018 was attributed to the measured increased concentration of particulate matter in the water column due to the resuspension of sediment and influx from the land, as well as simultaneous sediment redeposition on the leaves. A sentence linking the reduction of available light to the meadow with increased water turbidity in April 2018 has been added to Abstract. It now reads: The C. nodosa decline was most likely triggered in

April 2018 when light availability to the plant was drastically reduced. Such conditions resulted from increased seawater turbidity due to terrigenous input, sediment resuspension and elevated autotrophic biomass.

2. Decline of the seagrass meadow can hardly be attributed to light availability, grazing pressure or others. It must be a result of multiple drivers impact. Therefore, the discussion should be re-sorted and holistically analyzed. Authors response: We agree that the decline of the seagrass meadow in Saline Bay was the result of multiple drivers' impact. It all began with the increased water turbidity which weakened the plant and made it susceptible to other stressors such as lack of oxygen and H2S penetration within the plant tissue. Marks of visible grazing were only observed in July and August 2017 and not in April 2018 when the die-off was triggered. For this reason we did not include grazing as one of the main drivers in the decline of C. nodosa meadow. The discussion has been re-arranged accordingly. The discussion now reads: Saline bay is a shallow, highly dynamic coastal area characterized by frequent turbid waters due to the combined effect of land run-off and wind-driven resuspension of the fine sediment. Nutrients and Chl a (as a proxy for autotrophic biomass) varied in the ranges characteristic for the oligotrophic coastal waters off Rovinj (Ivančić et al., 2018). The increases in particulate matter concentration were associated with freshwater input, while their enrichment with unsaturated fatty acids, deriving from phytoplankton, was observed during the increases of autotrophic biomass. Only in September 2017, this increase was supported by nutrients from the water column, while all other less pronounced increases were most likely supplied with nutrients through sediment resuspension. In temperate Mediterranean coastal waters C. nodosa meadows show a clear unimodal annual growth cycle, reaching maximum development in summer, minima during winter and a particularly active phase in spring (Terrados and Ross, 1992; Zavodnik et al., 1998; Agostini et al., 2003). In Saline Bay, the maximum growth was shifted towards early autumn. This shift was most likely due to an intense grazing activities (Cebrian et al., 1996; Valentine and Duffy, 2006) suggested by a prevalence of visibly grazed leaves during July and August 2017. A minimum growth occurred during late autumn/winter,

as commonly observed. However, during the spring of 2018, phenological parameters continued to decrease despite established favorable environmental conditions for growth, i.e. increase in water temperature, intensity and period of solar radiation. This decrease continued until the complete extinction of the aboveground tissue in August 2018. The belowground tissue followed a similar trend, but with less expressed changes. Still, their recognizable remnants were found after the loss of above-ground tissues. Organic matter and closely correlated total lipids in the sediment of C. nodosa rooted area changed significantly throughout the investigated period, in contrast to the organic matter in non-vegetated sediment. Nevertheless, considerable similarity in the quality and degradation of lipid matter at both, the vegetated and the non-vegetated areas indicates an important contribution of detritus from the meadow as a source of organic matter for prokaryotes in non-vegetated sediments. This close coupling could be expected due to sites proximity and lower organic content of the non-vegetated sediment, which should enhance the dependence of prokaryotes on the imports of seagrass detritus from the adjacent meadows (Holmer et al., 2004). Significant enrichment of C. nodosa sediment with unsaturated, more labile components only during abundant growth of meadow could be explained by more efficient entrapment of seston material from the water column within the meadow (Gacia and Duarte, 2001). Such easily utilizable organic matter, including dissolved monomeric carbohydrates, leaching out during decomposition of C. nodosa leaves stimulates prokaryotic growth (Peduzzi and Herndl, 1991). This effect could be observed, as a higher prokaryotic abundance in C. nodosa sediment during this period. From July 2017 to March 2018, an adaptation of C. nodosa leaves to the decreasing light and temperature occurred. Until October 2017, the temperature of the water column was optimal for elongation of the leaves and biomass increase, while the ambient light intensities were continuously decreasing. An additional reduction of available light might occur from self-shading effect due to high canopy biomass and epiphytic macroalgae growth. Desaturation of low and relatively invariable lipids during the most active growth suggested the increase in the membrane fluidity to optimize photosynthetic activity under low light condition. Such physiological

adaptation was found in seagrasses along a depth gradient (Beca-Carretero et al., 2019) and macroalgae in contrasting seasons (Schmid et al., 2014). In late autumn the decrease in desaturation indicated a reduced fluidity and activity of photosynthetically active membranes. The lower fluidity reduces proton leakage through the thylakoid membranes and energy consumption for their maintenance (Quigg et al., 2006; Wacker et al., 2016). The reduced photosynthetic activity was associated with a decreased density of shoots and aboveground biomass. By shedding leaves and shoots the plant further balances metabolic requirements and mobilize energy from the carbohydrate reserves stored in the below-ground tissue (Alcoverro et al., 2001; Lee et al., 2007). During the winter, the rapid desaturation of increasing lipids could be attributed primarily to a sharp and continuous decrease in water temperature. The desaturation provides a mechanism for the thermo-adaptive regulation of membrane fluidity and cold resistance in algae and plants (Terrados and Lopezjimenez 1996; Iveša et al. 2004; Upchurch, 2008). In a healthy seagrass meadow, the oxygen generated by seagrass photosynthesis is transported to below-ground tissues to maintain an oxic microsphere around roots and rhizomes, where it oxidizes sulfide to non-toxic S0, preventing the intrusion of H2S into the plant and the anoxia of the roots (Pedersen et al., 1998; Holmer et al, 2005). S0 was found in the C. nodosa below-ground tissue during the entire investigated period, as regularly observed in seagrasses living in sulfidic sediments (Holmer et al. 2006; Holmer and Hasler-Sheetal, 2014; Hasler-Sheetal and Holmer, 2015). The relatively low accumulation of H2S (< 30 $\mu$M) during summer and early autumn 2017 indicated that H2S was apparently rapidly recycled within the rooted area via re-oxidation by O2 to S0 and/or removal by precipitation with iron compounds. Most of S0 was found in oxic layers or suboxic/anoxic boundaries with concentrations typical for sulfidic coastal sediments (Troelsen and Jørgensen, 1982; Panutrakul et al., 2001; Pjevac et al., 2014). The oxidation of H2S could occur spontaneously by chemical reaction with free oxygen or mediated by sulfide-oxidizing bacteria surrounding or being attached to seagrass roots (Jørgensen, 1977; Cucio et al., 2016; Ugarelli et al., 2017; Fahimipour et al., 2017). In November, due to the degradation of organic matter, reduced oxygen production and leakage in the rooted zone caused by C. nodosa senescence, the re-oxidation capacity of the sediment was greatly decreased. This resulted in considerable accumulation of H2S (> 100 $\mu$M) which extended up to the sediment surface. In winter and early spring, H2S production generally decreased, likely due to reduced activity of the sulfate-reducing bacteria at lower temperatures, and the sediment gradually shifted to a more oxidized state. H2S detected even in the oxic sediment and in the rooted area could be attributed to the sediment hetero-geneity and the presence of reducing micro-niches where anaerobic metabolism could occurs regardless of surrounding redox conditions (Jørgensen 1977; Frederiksen and Glud, 2006). In April 2018, C. nodosa had been most probably exposed to increased siltation, due to an increase in terrigenous input combined with resuspension of sedi-ment provoking elevated autotrophic growth. The intensive siltation is associated with the increased light attenuation, both through the direct shading effect of suspended sediments and through the promotion of phytoplankton and epiphyte growth by the as-sociated increase in nutrients (Terrados et al., 1998; Halun et al., 2002; Brodersen et al., 2015). Therefore, the increase in seawater turbidity and considerable sediment redeposition on the leaves might have been severely impaired the light availability and photosynthetic activity. The reduction of light below minimum requirements for C. no-dosa survival was indicated by LA/ALA > 1 in C. nodosa above-ground tissue due to a decrease in the conversion of LA to ALA (Harris and James, 1965). When mini-mum light requirements are not met ($\sim$14% of incidence light) the seagrass intensely sheds its leaves and shoots, while at a decrease to < 1% die-off is inevitable (Collier et al., 2012). This reduced light condition apparently persisted until May 2018 and most likely prevented the re-establishment of photosynthesis and C. nodosa continued to shed shoots and leaves. The reduced photosynthesis and therefore O2 transport from the leaves to the rhizome-root system probably minimized root respiration. The maintenance of the oxic rhizosphere and the internal O2 partial pressure in the lacunae further depended mainly on the diffusion of O2 from the water column (Pedersen et al., 1998; Greve et al., 2003; Sand-Jensen et al., 2005). From April to June 2018, O2 in the

bottom water drastically decreased. Due to poor supply, O2 content of below-ground tissue was too low to maintain an oxic microenvironment and therefore, the plant tissues became potentially accessible to sulfide intrusion (Pedersen et al., 2004). At the same time, the sediment was enriched with new organic matter derived from increased autotrophic biomass in bottom waters. In addition to the induction of the benthic bloom, strong sediment resuspension, most likely by aeration, stimulated the intense oxidation of H2S that started to produce in the rooted zone (up to 180 $\mu$M, Fig. 7). An increase in S0 concentration that reached its maximum in the same layer suggests a simultaneous oxidation of the produced H2S. The sulfide oxidation probably caused oxygen depletion in the rooted zone and anoxic zone extension up to the sediment subsurface. In May 2018, the excess of organic matter accumulated in April 2018 was degraded. The concentrations of S0, detected only in the suboxic layer, considerably decreased possibly by disproportionation or respiration by members of the sulfate-reducing bacteria. S0-disproportionating Desulfobulbaceae and S0-respiring Desulfuromonadales are frequently detected in anoxic coastal sediment (Pjevac et al., 2014). During June and July 2018, a sudden and significant deterioration of C. nodosa physiological condition was indicated by the increase in LA/ALA ratio in the leaves and overall saturation of decreasing lipids in above- and below-ground tissues. Additionally, the loss of leaf tissue negatively impacted the photosynthetic carbon fixation (Lee and Dunton, 1997; Lee et al., 2007). The below-ground tissue that was not supported by photosynthetically derived oxygen became anoxic. Thus induced anaerobiosis most likely caused a complete inhibition of the fatty acid desaturation chain (Harris and James, 1965) and permanent breakdown of photosynthesis. In July, the bottom waters were completely depleted in O2 and the whole plant exposed to sulfides. To reach the leaves, sulfide has to exceed below-ground tissue oxidation capacity, invading the meristems where sulfide toxicity can have drastic effects on shoot growth and survival (Greve et al., 2003; Frederiksen et al., 2008). H2S inhibit cytochrome c oxidase by binding to regulatory sites on the enzyme, reducing the rate of cellular respiration and leading to the chemical asphyxiation of cells (Nichols et al., 2013). From June to August 2018, the

decomposition of organic matter, encompassing the entire sediment core, was intensified and accompanied by a large increase in H2S concentrations (up to 1200 $\mu$M). The degradation process involved rhizomes and roots, as suggested by an apparent loss of below-ground biomass. Such loss typically occurs in the first stage of plant decay, the leaching phase (Trevathan-Tackett et al., 2017). Readily available, soluble carbohydrates that largely contribute to leachate mass (Vichkovitten and Holmer, 2004) most probably supported the increase in prokaryotic abundance observed in June and July 2018, and also high rates of sulfate reduction. However, the significant decreases in PA that coincided with a maximum degradation of organic matter and H2S production in August 2018 might indicate that remaining compounds were not degradable by the sulfate reduction pathway (Arndt et al., 2013) and needed the presence of prokaryotes specialized in the anaerobic degradation of refractory compounds, including cellulose and lignin. During September and October 2018, due to the combined effect of freshened oxygenated water input and resuspension which gradually deepen the oxic layer, H2S concentrations drastically decreased due to re-oxidation. Biogeochemical studies suggest most sulfide (80–90%) is eventually re-oxidized, while only 10–20% is ultimately buried as complexes with iron (i.e. FeS, FeS2) or with organic matter after sulfurization (Jørgensen, 1977; 1982). H2S scavenging with iron and formation of iron sulfides might be more critical in Saline Bay since terrestrial waters are washing out terra rossa, rich in Fe-oxides and oxyhydroxides (Durn, 2003). For this reason, sediment cores were most likely always black with sulfuric odor, irrespective of H2S concentrations or presence of vegetation.

3. Decreased root respiration may also contribute to the dying off Authors response: We agree that the decreased root respiration contributed to C. nodosa dying off. We believe that reduced O2 transport from the leaves to the rhizome-root system minimized or even stopped root respiration. Moreover, the lack of sediment oxygenation compromised the metabolic activities of aerobic bacteria around the roots and their oxidation of H2S. A commentary regarding root respiration was added within the rearranged Discussion.

4. Changes of unsaturated fatty acids could be attributed to many sources, since phytoplankton or microalgae are the main producers of these compounds, it is hard to guess. Authors response: We are aware that changes in unsaturated fatty acids could be attributed to many sources. Particularly in the sediment their changes depend on the dynamic interactions between primary producers and their consumers within the food web. In this paper, the intention was to compare vegetated and control (non-vegetated) sediment using different markers and indices which together provided an indicative interpretation of the predictable sources. More complex analysis of the sources exceeds the scope of this paper and will be the topic of further investigation.

5. While grazing rate might be responsible for the changes in seasonal change of stand crop of the seagrass, the authors did not provide any record that grazing rate is higher in the Saline Bay. Authors response: In July and August 2017 the leaves were very short with visible signs of bites, so we assumed that significantly lower biomass in these two months was the result of grazing activity of herbivores. In addition, in a nearby bay where we performed biometric measurements in C. nodosa meadow simultaneously, we did not notice such intense leaves damage and the vast majority of leaves had intact apexes.

6. During summer period, high light and temperature may synergistically reduce the biomass of the seagrass due to higher respiration and higher photoinhibition. Authors response: We are aware of this synergistic effect of high light and high temperature on the reduction of biomass related to seagrasses growing near or within the intertidal zone where they may be exposed to high light stress which may then result in down-regulation of photosynthetic apparatus or if irradiance is too high by photoinhibition. As the sampling depth in Saline Bay (3 - 4 m) was not in the intertidal region and tidal oscillation does not exceed 50 cm, we believe that this mechanism did not contribute to the reduction of C. nodosa biomass either in summer 2017 or in summer 2018. The biomass reduction and meadow die-off started when the temperature were still moderate. When the temperature reached its maximum (August 2018) the meadow had

already died and therefore, the effect of the high temperature could not be displayed.

Technical corrections: 1. Repeated wordings should be avoded in a sentence or paragraph. Authors response: This was checked and corrected accordingly.

2. Unit of silicate should be double checked, might be mistaken Authors response: The unit of silicate was double checked. Silicates (orthosilicates or reactive silicates) were determined spectrophotometrically by molybdenum blue method. Calibration was performed in the range 0.5 – 20 $\mu$mol L-1of sodium silicofluoride, Na2SiF6. Accordingly, the results were presented in $\mu$mol L-1 ($\mu$M).

Line 83 change shorter to shorten. . .changed

---

## Author Response (AR1)

*Dear Editor,*

*We greatly appreciate all of editor and reviewers' comments and suggestions which have been accepted in revised version of our manuscript: Dynamics of environmental conditions during a decline of a Cymodocea nodosa meadow (bg-2019-484). We believe we have satisfactorily addressed them.*

Associate Editor Decision: Reconsider after major revisions (29 Mar 2020) by Minhan Dai

Comments to the Author:

Re: g-2019-484 "Dynamics of environmental conditions during a decline of a Cymodocea nodosa meadow" by Mirjana Najdek et al.

Dear authors,

I went through both your MS and your interactive responses to the reviews. I encourage you to submit a thoroughly revised MS by carefully consider these reviews. In particular, I urge you to address the critical comments on light raised by both reviewers.

When you submit your revised MS, you need to provide a point-to-point letter explaining how you address the comments and concerns from the reviewers. Your revised MS will be sent out for further reviews.

Sincerely,

Minhan Dai

*Editor*

*Please, find below the short list of changes we made in our revised manuscript and a point-to-point letter explaining how we addressed the comments and concerns from the reviewers:*

- *In Abstract the sentence concerning light availability decrease in April 2018 was rewritten. The sentence about recovery of the below-ground tissue was rephrased.*
- *In Introduction the newer literature about preservation of marine diversity and carbon sequestration was provided.*
  *Introduction about seagrass meadows in Adriatic and research gap was added.*
- *In Materials and Methods a map was provided (now Fig S1). The sampling dates and depths of C. nodosa were included.*
- *The Discussion has been re-arranged.*
- *The Conclusion has been rewritten as advised.*
- *Figure 4…The years are annotated and an explanation is added in the legend*
- *Supplementary material…A sampling map was inserted as Fig S1.*

*Please find our point-by-point response to each reviewer's comments below:*
The authors investigated the dynamics of environmental conditions during a decline of a Cymodocea nodosa meadow in the northern Adriatic Sea, analyzed the correlation between those physicochemical and biological parameters, and concluded that the reduced light availability and following photosynthesis was the most likely reason leading to the decay of seagrass meadow. The experiments seem to be conducted carefully and the results were thoroughly discussed. This study supplies helpful information on understanding the decline of seagrass globally. However, there are some points that the authors need to attend to before it can be published in Biogeosciences.

I have two general concerns about this study.

**COMMENT**: The loss of seagrass meadow is attributed to reduced light availability and thus photosynthesis in this study. However, it seems that there is no direct data to support this conclusion. Have the light intensity in water column and photosynthetic rate of seagrasses were measured? The solar radiation in April should not be the lowest level compared to other months.

*RESPONSE***:** *Light intensity in the water column and photosynthetic rates of seagrass were not measured and there are no direct data to support this conclusion. However, indirectly from the data, the reduction of available light to the C. nodosa meadow in April 2018 was indicated by a visible increase in turbidity of the water column, due to an increased concentration of particulate matter of terrigenous origin and sediment resuspension. Terrigenous input in combination with sediment resuspension covered the meadow with sediment; thereby significantly reducing the amount of light accessible to the plant. Indeed, the solar radiation in April should not be at the lowest level compared to other months, therefore we would rule out the decrease in solar radiation as a potential trigger of the seagrass decay in April 2018.*

**COMMENT**: The authors mentioned that from July 2017 to March 2018, C. nodosa successfully adapted to the changes of environmental conditions and prevented H2S accumulation by its re-oxidation, supplying the sediment with O2 from the water column and/or leaf photosynthesis. Then why did not C. nodosa adapt to the environmental changes from April 2018 onwards? I am wondering that the decline of seagrass meadow in the northern Adriatic Sea is a natural process or caused by other drivers?

*RESPONSE***:** *We believe that already in April 2018, C. nodosa received the amount of light below the minimum required for photosynthesis, as suggested by an increase in LA/ALA ratio > 1 (as conversion of LA to ALA declines in dark being completely inhibited by anaerobiosis). Besides reduced photosynthesis and therefore supply of the below-ground tissue with oxygen, a depletion of oxygen due to intense oxidation of $H_2S$ occurred in the sediment, thus creating anoxic conditions in most of the rooted areas. Most likely, this change in April 2018 drastically and irreversibly compromised the survival of the meadow. From April 2018 onwards, C. nodosa didn't reestablish photosynthesis and continued to lose shoots and overall biomass, while at the same time in the sediment, the concentration of $H_2S$ increased considerably, which, due to lack of oxygen, penetrated the plant and caused the meadow die-off. We would exclude that the process in Saline Bay was natural, because several geographically nearby meadows didn't display a similar pattern of regression.*

**COMMENT**: Line 22 Why did light availability decrease in April 2018?

**RESPONSE**: *The available light to the C. nodosa was reduced due to increased seawater turbidity resulted from combined effect of terrigenous input, sediment resuspension and elevated autotrophic biomass. The sentence was rewritten and amended.*

**CHANGE (Page 2, Lines 23 – 26)**: *The C. nodosa decline was most likely triggered in April 2018 when light availability to the plant was drastically reduced. Such conditions resulted from increased seawater turbidity due to terrigenous input, sediment resuspension and elevated autotrophic biomass.*

**COMMENT:** Line 30 The data in Figure 2 did not show the recovery of the below-ground tissue?

**RESPONSE**: *We agree with this comment. The sentence in the Abstract was rephrased.*

**CHANGE (Page 2, Lines 33 – 34):** *The influx of oxygenated waters in September 2018 led to the re-establishment of $H_2S$ oxidation in the sediment and the remaining of the below-ground tissue.*

**COMMENT** Line 37 Better to supply latest literature as there are loads?

**RESPONSE**: *The following literature has been added*

**CHANGE (Page 3, Lines 42 – 43):** *Duarte, C.M., Kennedy, H., Marbà, N., Gacia, E., Fourqurean, J.W., Beggins, J., Barrón, C., Apostolaki, E.T.: Seagrass community metabolism: Assessing the capacity of seagrass meadows for carbon burial: Current limitations and future strategies. Ocean Coast. Manag., 83, 32-38, 2013. Samper-Villarreal, J., Lovelock, C.E., Saunders, M.I., Roelfsema, C., and Mumby, P.J.: Organic carbon in seagrass sediment is influenced by seagrass canopy complexity, turbidity, wave height, and water depth. Limnol. Oceanogr., 61, 938-952, 2016.*

**COMMENT**: Line 41 Add a comma after matter… *a comma was added*.

**COMMENT**: Line 85 A introduction about seagrass meadows in Saline Bay or Adriatic Sea should be supplied here. Meanwhile, it would be helpful to add research gap here.

**RESPONSE**: *Introduction about seagrass meadows in Adriatic and research gap was added.*

**CHANGE (Page 4, Lines 90 – 100):** *The seagrass Cymodocea nodosa (Ucria) Ascherson is widely distributed and common species throughout the Mediterranean (Terrados and Ros 1992; Pedersen et al., 1997; Cancemi et al., 2002; Agostini et al., 2003). For the northern Adriatic, however, only sparse data are available on the standing crop, seasonal dynamics or natural/anthropogenic pressures supporting the ecological or conservation status of C. nodosa meadows (Zavodnik et al., 1998; Orlando-Bonaca et al., 2015). Although C. nodosa show large phenotypic plasticity adapting to diverse natural and anthropogenic stressors by physiological and morphological adaptations, a severe decline has been reported during the last decades in coastal areas (Orth et al., 2006; Short et al., 2011; Tuya et al., 2002; 2014), including the northern Adriatic (Orlando-Bonaca et al., 2015; 2019). One of these declines is documented in our study performed in Saline Bay (northern Adriatic Sea) from July 2017 to October 2018.*

*New references: Orlando-Bonaca, M., Francé, J., Mavrič, B., Grego, M., Lipej, L., Flander Putrle, V., Šiško, M., and Falace, A.: A new index (MediSkew) for the assessment of the Cymodocea nodosa (Ucria) Ascherson meadow's status. Mar. Environ. Res., 110, 132-141, 2015.*

*Orlando-Bonaca, M., Francé, J., Mavrič, B., and Lipej, L.: Impact of the Port of Koper on Cymodocea nodosa meadow. Annales, 29, 187-194, 2019.*

*Short, F.T., Polidoro, B., Livingstone, S.R., Carpenter, K.E., Bandeira, S., Bujang, J.S., Calumpong, H.P., Carruthers, T.J.B., Coles, R.G., Dennison, W.C., Erftemeijer, P.L.A., Fortes, M.D., Freeman, A.S., Jagtap, T.G., Kamal, A.M., Kendrick, G.A., Kenworthy, W.J., La Nafie, Y.A., Nasution, I.M., Orth, R.J., Prathep, A., Sanciangco, J.C., van Tussenbroek, B., and Vergara, S.G.: Extinction risk assessment of the world's seagrass species. Biol. Conserv., 144, 1961-1971, 2011.*

*Tuya, F., Martín, J.A., and Luque, A.: Impact of a marina construction on seagrass bed at Lanzarote (Canary Islands). J. Coast. Conserv., 8, 157-162, 2002.*

*Tuya, F., Ribeiro-Leite, L., Arto-Cuesta, N., Coca, J., Haroun, R., and Espino, F.: Decadal changes in the structure of Cymodocea nodosa seagrass meadows: Natural vs. human influences. Estuar. Coast. Shelf Sci., 137, 41-49 (2014).*

**COMMENT**: Line 98 Better to supply a map for the study site.

**RESPONSE**: *A map is now added as Fig S1 **(Page 5, Line 110)** and provided in Supplementary material **(Page 5)***

**COMMENT**: Line 116 Sampling time/frequency needs to be stated. What is the depth for C. nodosa living?.

**RESPONSE**: *The sampling dates and depths of C. nodosa were included*

**CHANGE (Page 5, Line 126 and 128):** *The sampling was performed for 15 months from July 2017 to October 2018. C. nodosa (3 - 4 m of depth) was collected together with rhizomes, roots and macroalgae by divers using the quadrat sampling method.….*

**COMMENT**: Line 320 This is true as shown in the green macroalgae Ulva linza (Gao et al. 2018 Food Chemistry, 2018, 258: 71-78).?

**RESPONSE**: *The reference was added in the text **(Page 12, Line 331)** and listed in References.*

**COMMENT**: Line 386 What are these prokaryotic organism?

**RESPONSE**: *Prokaryotic organisms are Bacteria and Archaea from 0.2 – 2 μm, stained with DAPI and counted by epifluorescence microscopy.*

**COMMENT**: Lines 485-487 How did you know it? Any data or literature to support this speculation??

**RESPONSE**: *In July and August 2017 the leaves were short with visible signs of bites, so we assumed that significantly lower biomass in these two months was the result of grazing*

*activity of herbivores. Generally, the meadows have been shown to be an important source of food for herbivores. We rephrased the sentences to avoid misleading interpretation and have added the references that support biomass loss by herbivory in C. nodosa.*

***CHANGE (Page 17, Lines 501 – 504)**: In Saline Bay, the maximum biomass was measured in October 2017. This shift from summer to early autumn was most likely due to an intense grazing activities (Cebrian et al., 1996; Valentine and Duffy, 2006) suggested by a prevalence of visibly grazed leaves during July and August.*

*New references: Cebrian, J., Duarte, C.M., and Marbà, N.: Herbivory on the seagrass Cymodocea nodosa (Ucria) Ascherson in contrasting Spanish Mediterranean habitats. J. Exp. Mar. Biol. Ecol., 204, 103-111, 1996*

*Valentine, J.F., and Duffy, J.E.: The central role of grazing in seagrass ecology. In: Seagrasses: Biology, Ecology and Conservation, Springer, Netherlands, pp 431-501, 2006.*

**COMMENT:** Line 680 Conclusion should not be a repetition of Abstract. The purpose of a conclusion is to tie together, or integrate the various issues, findings, arguments etc., covered in the body of the paper, and to make comments upon the meaning of all of it. This includes noting any implications resulting from your discussion of the topic, as well as recommendations, forecasting future trends, and the need for further research.

***RESPONSE**: The Conclusion has been rewritten.*

***CHANGE (Page 24, Lines 717 – 729)**: Our results provide insights into the interaction of multiple stressors that have led to the meadow decay, triggered in the sensitive recruitment phase of meadow growth. Even after the improvement of the sediment conditions by the end of the summer 2018, C. nodosa was not able to recolonize its previously occupied areas. This finding combined with a visible alteration of the water column and sediment indicates a considerable loss of the C. nodosa habitat. Further research is needed to examine the fate of Saline Bay meadows and an eventual recolonization of the area. Beyond seagrass itself, this loss had extensive consequences as it has endangered many species that depend on seagrass for food, shelter and nursery. Given the lack of data on the ecological and conservation status of the still numerous seagrass meadows along the northern Adriatic coast, the identification and monitoring of the main pressures acting on them are needed to protect such valuable habitats from degradation and extinction.*

**COMMENT** Line 950 Please annotate which year for the months and explain why it ends in Feb in the legends.

***RESPONSE**: The years are annotated and an explanation is added in the Figure 4 legend.*

***CHANGE (Page 39, Line 1012)**: Figure 4. The contribution of macroalgal phyla in a meadow and total macroalgal biomass; after February 2018 macroalgae were no longer present in the C. nodosa meadow.*
The paper entitled "dynamics of environmental conditions during a decline of cymodocean nodosa meadow" reported biomass changes of the seagrass along with environmental changes in both seawater and sediments during one year period of 2017 to 2018. The results showed that C. nodosa successfully adapted to the changes of environmental conditions and prevented H2S accumulation by its re-oxidation, supplying the sediment with O2 from the water column and/or leaf photosynthesis, implying that the C. nodosa die-off would be most likely caused in April 2018 by a reduction of light availability. Unfortunately, solar irradiances changes either at the surface of water column or in water were not supplied. Especially, the light levels during investigation periods were not provided. Attenuation of light in water during different seasons with different amounts of PM can be directly link to photosynthetic performance of the seagrass. While the authors showed that in the sediment, depletion of oxygen due to intense oxidation of H2S led to anoxic conditions in most of the rooted areas. This could negatively affect respiration of the plant root, therefore, its growth. With reduced growth capacity, high concentrations of H2S were observed in the sediment cores and bottom waters. This is an interesting result indicating the relationship of H2S levels with photosynthetic O2 evolution. Generally speaking, the paper has scientific significance, and is suitable to be accepted to BG after revision.

Specific comments:

**COMMENT**: Daily sunlight doses should be in parallel with water temperature, and should be provided, which can be easily obtained from local meteorological stations if the authors did not measure.

**RESPONSE**: *We are aware that the temperature of natural water bodies varies in response to diurnal and seasonal changes in solar radiation. Our temperature data showed no unusual diurnal and/or seasonal pattern. Therefore, it could be reasonably assumed that during the study period there was no big difference in the amount of solar radiation received by the sea. Also we believe that solar radiation data from a meteorological station would not help here. From reporting solar radiation measured in the lower atmosphere (as typically done by meteo-stations), the solar radiation reaching the seagrass meadow in this coastal system cannot be extrapolated. Underwater solar radiation measurements would be required which, unfortunately, we don't have. The penetration of light in the water column, which regulates photosynthesis, is strongly influenced by the transparency of the water. The attenuation of light in the water column and therefore, the reduced availability of light required for C. nodosa photosynthesis in April 2018 was attributed to the measured increased concentration of particulate matter in the water column due to the resuspension of sediment and influx from the land, as well as simultaneous sediment redeposition on the leaves. A sentence linking the reduction of available light to the meadow with increased water turbidity in April 2018 has been added to Abstract.*

**CHANGE (Page 2, Lines 23 – 26):** *The C. nodosa decline was most likely triggered in April 2018 when light availability to the plant was drastically reduced due to increased seawater turbidity that resulted from increased terrigenous input combined with resuspension of sediment and elevated autotrophic biomass.*

**COMMENT:** Decline of the seagrass meadow can hardly be attributed to light availability, grazing pressure or others. It must be a result of multiple drivers impact. Therefore, the discussion should be re-sorted and holistically analyzed.

*RESPONSE: We agree that the decline of the seagrass meadow in Saline Bay was the result of multiple drivers' impact. It all began with the increased water turbidity which weakened the plant and made it susceptible to other stressors such as lack of oxygen and H2S penetration within the plant tissue. Marks of visible grazing were only observed in July and August 2017 and not in April 2018 when the die-off was triggered. For this reason we did not include grazing as one of the main drivers in the decline of C. nodosa meadow. The Discussion has been re-arranged according to the periods of regular growth of the C. nodosa meadow (July 2017 – March 2018) and its decay (April-August 2018). We believe that in this way we presented more clearly the meadow decline as a result of multiple drivers' cascading impact. Also, in this way the repeated wordings in the sentences and paragraphs were avoided, as recommended, and this resulted in shorter discussion.*

[revised manuscript text omitted]
 input and resuspension which gradually deepen the oxic layer, $H_2S$ concentrations drastically decreased due to re-oxidation. Biogeochemical studies suggest most sulfide (80–90%) is eventually re-oxidized, while only 10–20% is ultimately buried as complexes with iron (i.e. FeS, $FeS_2$) or with organic matter after sulfurization (Jørgensen, 1977; 1982). $H_2S$ scavenging with iron and formation of iron sulfides might be more critical in Saline Bay since terrestrial waters are washing out terra rossa, rich in Fe-oxides and oxyhydroxides (Durn, 2003). For this reason, sediment cores were most likely always black with sulfuric odor, irrespective of $H_2S$ concentrations or presence of vegetation.*

**COMMENT**: Decreased root respiration may also contribute to the dying off

**RESPONSE:** *We agree that the decreased root respiration contributed to C. nodosa dying off. We believe that reduced $O_2$ transport from the leaves to the rhizome-root system minimized or even stopped root respiration. Moreover, the lack of sediment oxygenation compromised the metabolic activities of aerobic bacteria around the roots and their oxidation of $H_2S$. A commentary regarding root respiration was added within the re-arranged Discussion.*

**CHANGE (Page 21, Line 635 – 637)**: *The reduced photosynthesis and therefore $O_2$ transport from the leaves to the rhizome-root system probably minimized root respiration*

**COMMENT**: Changes of unsaturated fatty acids could be attributed to many sources, since phytoplankton or microalgae are the main producers of these compounds, it is hard to guess.

**RESPONSE**: *We are aware that changes in unsaturated fatty acids could be attributed to many sources. Particularly in the sediment their changes depend on the dynamic interactions between primary producers and their consumers within the food web. In this paper, the intention was to compare vegetated and control (non-vegetated) sediment using different markers and indices which together provided an indicative interpretation of the predictable sources. More complex analysis of the sources exceeds the scope of this paper and will be the topic of further investigation.*

**COMMENT**: While grazing rate might be responsible for the changes in seasonal change of stand crop of the seagrass, the authors did not provide any record that grazing rate is higher in the Saline Bay.

**RESPONSE**: *In July and August 2017 the leaves were very short with visible signs of bites, so we assumed that lower biomass and leaf length in these two months were the result of grazing activity of herbivores. In addition, in a nearby bay where we performed biometric measurements in C. nodosa meadow simultaneously, we did not notice such intense leaves damage and the vast majority of leaves had intact apexes. We rephrased the sentences to avoid misleading interpretation and have added the references that support biomass loss by herbivory in C. nodosa.*

**CHANGE (Page 17, Lines 501 – 504)**: *In Saline Bay, the maximum biomass was measured in October 2017. This shift from summer to early autumn was most likely due to an intense*

*grazing activities (Cebrian et al., 1996; Valentine and Duffy, 2006) suggested by a prevalence of visibly grazed leaves during July and August.*

**COMMENT**: During summer period, high light and temperature may synergistically reduce the biomass of the seagrass due to higher respiration and higher photoinhibition.

*RESPONSE: We are aware of this synergistic effect of high light and high temperature on the reduction of biomass related to seagrasses growing near or within the intertidal zone where they may be exposed to high light stress which may then result in down-regulation of photosynthetic apparatus or if irradiance is too high by photoinhibition. As the sampling depth in Saline Bay (3 - 4 m) was not in the intertidal region and tidal oscillation does not exceed 50 cm, we believe that this mechanism did not contribute to the reduction of C. nodosa biomass either in summer 2017 or in summer 2018. The biomass reduction and meadow die-off started when the temperature were still moderate. When the temperature reached its maximum (August 2018) the meadow had already died and therefore, the effect of the high temperature could not be displayed.*

**COMMENT**: Repeated wordings should be avoided in a sentence or paragraph.

*RESPONSE: This was considerably avoided by rearranging the Discussion, otherwise checked and corrected accordingly throughout the text.*

**COMMENT**: Unit of silicate should be double checked, might be mistaken

*RESPONSE: The unit of silicate was double checked. Silicates (orthosilicates or reactive silicates) were determined spectrophotometrically by molybdenum blue method. Calibration was performed in the range 0.5 – 20 $\mu mol\ L^{-1}$ of sodium silicofluoride, $Na_2SiF_6$. Accordingly, the results were presented in $\mu mol\ L^{-1}$ ($\mu M$).*

**COMMENT**: Line 83 change shorter to shorten…*changed (Line 88)*

[revised manuscript text omitted]

---

## Author Response (AR2)

*Dear Editor,*

*We greatly appreciate all of editor and reviewers' comments and suggestions which have been accepted in revised version of our manuscript: Dynamics of environmental conditions during a decline of a Cymodocea nodosa meadow (bg-2019-484). We believe we have satisfactorily addressed them.*

Associate Editor Decision: Publish subject to minor revisions (review by editor) (19 May 2020) by Minhan Dai

Comments to the Author:

Dear Authors,

Thank you for submitting your revised MS, which has been further reviewed by two reviewers who had previously looked at your MS. Your paper is now potentially acceptable for publication at Biogeosciences with some additional discussion referring to the comment by reviewer #1 with respect to multiple stressors.

I look forward to seeing your further revised MS.

Sincerely,

Minhan Dai

Editor

*Please find below our point-by-point response to reviewer #1comments*:

**COMMENT:** In terms of the underwater light conditions, the authors argued, and responded with the edited sentence as follows " *C. nodosa* decline was most likely triggered in April 2018 when light availability to the plant was drastically reduced due to increased seawater turbidity that resulted from increased terrigenous input combined with resuspension of sediment and elevated autotrophic biomass."

Increased terrigenous input and reduced transparency should be supported with the data In fig. 1, in which the PM increased and salinity decreased, supporting the authors' argument.

*RESPONSE***:** The Abstract and discussion were slightly modified in order to better present the evidence supporting our arguments. Specific data were mentioned with reference to Fig.1b.

*CHANGE (page 2, lines 21 – 25):* The amended sentence in the Abstract now reads: The *C. nodosa* decline was most likely triggered in April 2018 when light availability to the plant was drastically reduced due to increased seawater turbidity that resulted from increased terrigenous input, indicated by a decrease in salinity accompanied with a substantial increase in particulate matter concentration, combined with resuspension of sediment and elevated autotrophic biomass.

*CHANGE (page 19, lines 559 - 563)*: The complementing sentence in the Discussion now reads: In April 2018, *C. nodosa* had been most probably exposed to increased siltation, due to an intensification of terrigenous input as indicated by a decrease in salinity ($\Delta$ 1.5 with respect to March) and a substantial increase in particulate matter concentration (up to 3 times higher than in March, Fig. 1b) combined with resuspension of sediment, provoking an elevated autotrophic growth.

**COMMENT**: In terms of multiple drivers impact, the authors responded as "It all began with the increased water turbidity which weakened the plant and made it susceptible to other stressors such as lack of oxygen and H2S penetration within the plant tissue"

Well, reduced availability or lack of O2 usually helps photosynthetic performance, since it affects the competition of oxygenation and carboxylation catalyzed by Rubisco. This has been evidenced in seagrass as follows:

Kim, M., Brodersen, K. E., Szabó, M., Larkum, A. W. D., Raven, J. A., Ralph, P. J., & Pernice, M. (2018). Low oxygen affects photophysiology and the level of expression of two-carbon metabolism genes in the seagrass Zostera muelleri. Photosynth Res., 136, 147-160. https://doi.org/10.1007/s11120-017-0452-1.

*RESPONSE*: We acknowledge the value of the comment regarding the effect of reduced oxygen availability on photosynthetic performance as presented by Kim et al. (2018) where they report that the net photosynthetic rate increased in response to reduced $O_2$ concentration in water. In our study, during the critical phase for the plant (in April 2018), the bottom waters was fairly oxygenated ($O_2$~125 μM, Fig. 6); therefore it can be assumed that an increased photosynthetic performance was not stimulated as shown under experimental conditions ($O_2$~ 8 μM) by Kim et al. (2018). What we meant by a lack of oxygen was referring to the oxygen levels within the plant tissue since the photosynthetic activity was probably suppressed by the sediment coating of the leaves and the overall high turbidity of the water column. Since from April to June 2018, $O_2$ in the bottom water drastically decreased, the sentence in discussion was added in order to compare our observations with the study presented by Kim et al., (2018). The increased photosynthetic activity with the reduction of light and oxygen availability in the water column might have occurred in late spring but was not sufficient to compensate the oxygen requirement of the plant tissues. This oxygen deficiency combined with an intrusion of $H_2S$ resulted in *C. nodosa* die-off.

*CHANGE (page 20, lines 578 – 582):* The added sentence now reads: Although in such conditions of limited light and $O_2$ the seagrass might be capable for rapid modulation of metabolic pathways and enhance the photosynthetic rate, as shown for *Zostera muelleri* (Kim et al., 2018), it appeared that $O_2$ content of the *C. nodosa* below-ground tissue was still too low to maintain the internal pressure and therefore, the plant tissues became potentially accessible to sulfide intrusion (Pedersen et al., 2004). The reference Kim et al. is added *(page 26, lines 770 -772)*.